# Bis-naphthopyrone pigments protect filamentous ascomycetes from a wide range of predators

Yang Xu[1], Maria Vinas [1,2], Albatol Alsarrag[1], Ling Su[1], Katharina Pfohl[1], Marko Rohlfs[3], Wilhelm Schäfer[4], Wei Chen[5] & Petr Karlovsky [1]

It is thought that fungi protect themselves from predation by the production of compounds that are toxic to soil-dwelling animals. Here, we show that a nontoxic pigment, the bis-naphthopyrone aurofusarin, protects *Fusarium* fungi from a wide range of animal predators. We find that springtails (primitive hexapods), woodlice (crustaceans), and mealworms (insects) prefer feeding on fungi with disrupted aurofusarin synthesis, and mealworms and springtails are repelled by wheat flour amended with the fungal bis-naphthopyrones aurofusarin, viomellein, or xanthomegnin. Predation stimulates aurofusarin synthesis in several *Fusarium* species and viomellein synthesis in *Aspergillus ochraceus*. Aurofusarin displays low toxicity in mealworms, springtails, isopods, *Drosophila*, and insect cells, contradicting the common view that fungal defence metabolites are toxic. Our results indicate that bis-naphthopyrones are defence compounds that protect filamentous ascomycetes from predators through a mechanism that does not involve toxicity.

[1] University of Goettingen, Molecular Phytopathology and Mycotoxin Research, 37077 Göttingen, Germany. [2] CIGRAS, University of Costa Rica, 2060 San Pedro, Costa Rica. [3] University of Bremen, Institute of Ecology, Population and Evolutionary Ecology Group, 28359 Bremen, Germany. [4] University of Hamburg, Biocenter Klein Flottbek, Molecular Phytopathology and Genetics, 22609 Hamburg, Germany. [5] Zhejiang University, College of Biosystems Engineering and Food Science, Department of Food Science and Nutrition, Hangzhou 310058, P.R. China. Correspondence and requests for materials should be addressed to W.C. (email: zjuchenwei@zju.edu.cn) or to P.K. (email: pkarlov@gwdg.de)

Soil fungi play a key role in nutrient cycling by degrading recalcitrant plant biomass. Fungal biomass is an attractive source of nutrients for soil invertebrates[1], and predation on fungi disrupts fungal networks[2,3] and modulates the composition[4] and activity[5] of fungal communities, thereby affecting fungal ecosystem services[6]. Because fungi are sessile organisms, their protection from predation consists primarily of chemical defence.

This chemical defence can be mediated by proteins or secondary metabolites. The role of fungal ribosome-inactivating proteins[7], protease inhibitors[8], and lectins[9,10] in fungal chemical defence has been elucidated at the molecular level. Studies of fungal defence metabolites have a long history, albeit with inconclusive outcomes. In 1977, Daniel Janzen suggested that fungal toxins protect moulded material from consumption by large animals and hinted that the same metabolites may protect infected grain from storage pests[11]. Janzen's ideas led to the hypothesis that mycotoxins protect fungi from predators, and the insecticidal properties of many mycotoxins have since been studied[12–14]. Apart from their toxicity to insects, circumstantial support for the role of mycotoxins in defence against predators has been drawn from the stimulation of mycotoxin synthesis by arthropod grazing[15] and mechanical injury[16] and from the accumulation of toxic metabolites in fungal reproductive

organs[17]. Although the ecological function of toxins accumulating in mushrooms (fruiting bodies of basidiomycetes) has been elucidated[18], efforts to substantiate the function of major mycotoxins of filamentous ascomycetes in their defence against predators have remained inconclusive[19,20]. Mycotoxin gliotoxin facilitates the escape of *Aspergillus flavus* during phagocytosis by a soil amoeba[21]; however, whether gliotoxin protects its producers from animal predators remains unknown. Two polyketides that have not been determined to be mycotoxins have been shown to protect two ascomycetes fungi from animal predation: Asparasone has protected the sclerotia of *Aspergillus flavus* from sap beetles[22], and neurosporin A has protected *Neurospora crassa* from springtail grazing[23]. Nevertheless, there is no indication that these findings can be generalised to related metabolites, other fungal species, or additional predators.

In this work, we investigate the effect of springtail grazing on the transcriptome of the filamentous ascomycete *Fusarium graminearum* (*F. graminearum*). The biosynthesis pathways for several secondary metabolites are induced via grazing. One of these metabolites is aurofusarin, which belongs to bis-naphthopyrones that are produced by many ascomycetes. Predation and mechanical damage stimulate aurofusarin synthesis. When mutants of *F. graminearum* with disrupted aurofusarin synthesis are offered to springtails, isopods, and mealworms, all predators strongly prefer the mutants over the aurofusarin-producing strains. Food choice experiments with purified aurofusarin, xanthomegnin, and viomellein—which are bis-naphthopyrones produced by many species of *Fusarium*, *Aspergillus* and *Penicillium*—reveal antifeedant effects of all three metabolites in mealworms and springtails. Toxicity assays with mealworms, springtails, *Drosophila* larvae, and insect cell cultures show a low toxicity of aurofusarin to arthropods. These results suggest that fungal bis-naphthopyrone pigments—which are widespread among ascomycetes—protect fungi from predators by exerting antifeedant effects on a wide range of phylogenetically distant arthropods.

## Results

**Predation stimulates the synthesis of aurofusarin.** Assuming that defence metabolites are synthesised on demand, we sequenced the transcriptome of the fungus *F. graminearum* that had been exposed to the springtail *Folsomia candida* to reveal which biosynthetic pathways were induced by grazing. RNA was extracted from fungal cultures after grazing, and the mRNA levels of all genes were determined by sequencing (RNAseq). Grazing was found to stimulate the transcription of pathways for the metabolites aurofusarin, fusarin C, and fusaristatin A (Fig. 1 and Supplementary Fig. 1b, c), additional gene clusters that are putatively involved in secondary metabolism (Supplementary Fig. 2), and seven genes that encode small proteins (Supplementary Fig. 1a). Pathways for the mycotoxins deoxynivalenol and zearalenone—which are toxic to insects[13,14]—and for necrosis and ethylene-inducing peptide-like proteins—which we hypothesised to be defence agents owing to their similarity to lectins[24]—were not induced by grazing. The RNAseq data are accessible under E-MTAB-6939 at ArrayExpress, EMBL–EBI (www.ebi.ac.uk/arrayexpress), and their analysis for 13,710 genes of *F. graminearum* can be found in Supplementary Data 1.

Aurofusarin was selected for further work because it is produced by many fungal species[25,26] and because metabolites of similar structures are produced by many genera of ascomycetes[27] (see below). Aurofusarin is a red pigment known from maize ears infected with *F. graminearum* (Fig. 2a) and pure cultures of the fungus (Fig. 2b). It belongs to dimeric naphtho-γ-pyrones (Fig. 2f). Springtail grazing stimulated the transcription

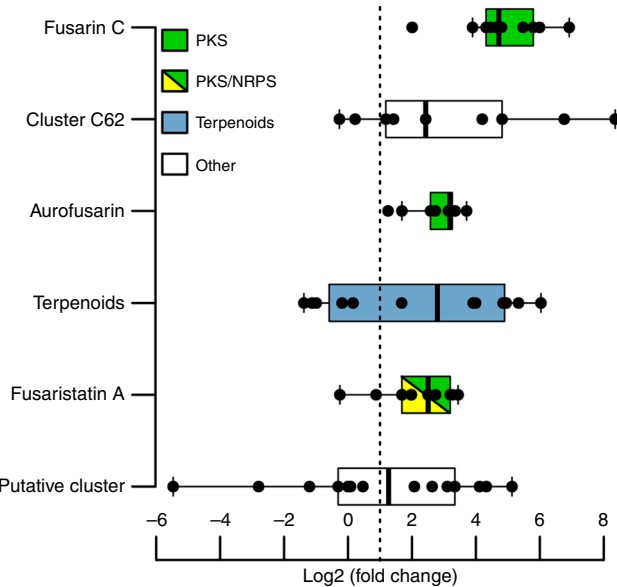

**Fig. 1** Secondary metabolite pathways upregulated by grazing in *F. graminearum*. *Fusarium graminearum* IFA66 was exposed to the springtail *Folsomia candida* for 48 h. RNA was extracted, and the levels of individual mRNAs were determined by RNA sequencing (RNAseq). Black points show $\log_2$(fold change) values for each gene in grazed versus control cultures. Upregulated gene clusters are defined as having > 50% of all genes and/or having the gene that encodes a signature enzyme be significantly induced (the $\log_2$(FPKM) was higher than 1.0 (dotted line), and the *q* value was lower than 0.01). Accession numbers: fusarin C (FGSG_07798, 07800–07805, and 13222–13224), cluster C62 (FGSG_10606, 10608, 10609, and 10611–10614, 10616, and 10617), aurofusarin (FGSG_02320–02329), terpenoids (FGSG_01737–01749), fusaristatin A (FGSG_08204–08210, 08213, and 08214), and putative cluster (FGSG_10557–10560, 10562–10567, 10569–10571, and 10573). Four biological replicates were used. Box plots show the median and interquartile range. Whiskers indicate the largest and smallest observation or 1.5-fold of the interquartile range, whichever is smaller or larger, respectively (Q1–1.5 × (Q3–Q1) or Q3 + 1.5 × (Q3–Q1)). Source data are provided in a Source Data file

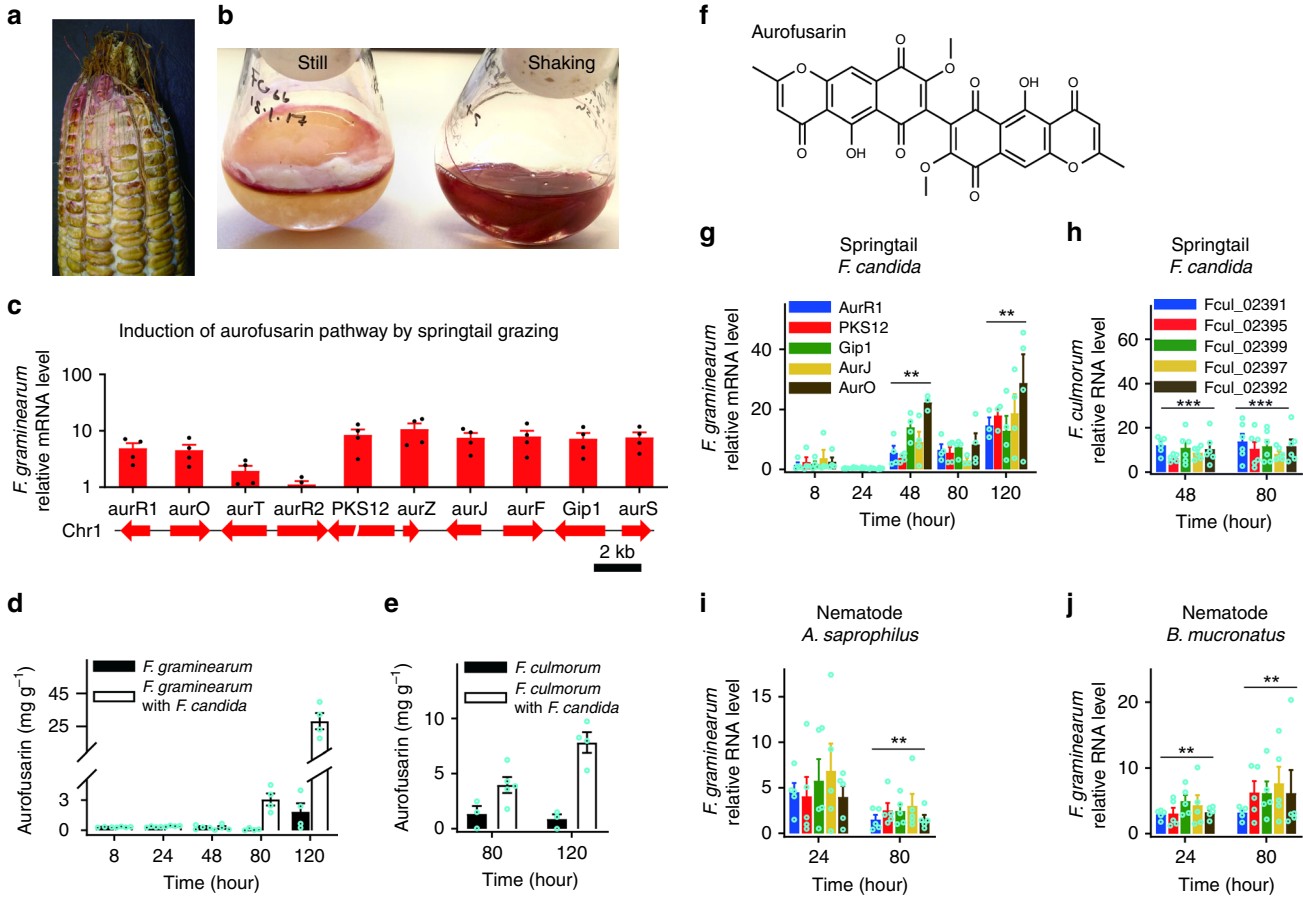

**Fig. 2** Aurofusarin synthesis in *Fusarium spp.* is stimulated by predation. **a** An ear of corn inoculated with *F. graminearum* showing red pigment aurofusarin (courtesy of Dr Belinda J. van Rensburg, ARC South Africa). **b** *F. graminearum* cultures in potato broth. **c** Upregulation of genes of aurofusarin biosynthesis in *F. graminearum* after exposure to grazing by the springtail *Folsomia candida* for 48 h (RNAseq; $n = 4$; see Fig. 1 for details). **d**, **e** Aurofusarin accumulation in *F. graminearum* and *F. culmorum* exposed to grazing by the springtail *Folsomia candida* ($n = 4$). **f** Structure of aurofusarin. **g**, **h** Upregulation of genes of aurofusarin synthesis in *F. graminearum* and *F. culmorum* after grazing by *Folsomia candida* (RT qPCR). **i**, **j** Upregulation of genes of aurofusarin synthesis after exposure to the fungivorous nematodes *Aphelenchoides saprophilus* and *Bursaphelenchus mucronatus* (RT qPCR). The gene cluster was labelled as significantly induced when mRNA levels of at least three genes increased at least threefold and the increase was statistically significant (**$P < 0.001$, ***$P < 0.0001$, two-tailed $t$ test) with both reference genes (glyceraldehyde-3-phosphate dehydrogenase and elongation factor 1a). Error bars show s.e.m. Three to four biological replicates were used in RT qPCR (see Supplementary Data 3). Source data are provided in a Source Data file

of all genes of the aurofusarin cluster except one (Fig. 2c). To examine whether aurofusarin synthesis was also induced by other predators, *F. graminearum* and *F. culmorum* were subjected to feeding by the springtail *F. candida* and the nematodes *Aphelenchoides saprophilus* and *Bursaphelenchus mucronatus* for different time periods, and relative mRNA levels for five genes of aurofusarin synthesis[28] were estimated by RT qPCR (Fig. 2g–j). Predation induced the aurofusarin pathway in all fungus/animal combinations. The estimation of the aurofusarin concentration in *F. graminearum* and *F. culmorum* cultures that had been subjected to springtail grazing by high-performance liquid chromatography (HPLC) with light absorption detection (HPLC-DAD) revealed that aurofusarin accumulation was simulated by grazing and that aurofusarin in grazed mycelia amounted to up to 2.5% of the dry weight (Fig. 2d, e). We were not aware of any non-polymeric secondary metabolite that accumulates in fungal mycelia at such a level, and we therefore determined the aurofusarin content in the mycelia of five *Fusarium* species grown in liquid cultures by HPLC with mass spectrometric detection (HPLC-MS/MS) (Supplementary Fig. 3). Aurofusarin levels of 1–7% of dry weight were found in four *Fusarium* species. Because both HPLC-DAD and HPLC-MS rely on aurofusarin standards, which are notoriously unstable

(see Methods), extracts of six *F. venenatum* cultures were re-analysed via HPLC with evaporative light-scattering detection (ELSD) for additional verification. ELSD is less accurate than DAD or MS yet does not require aurofusarin standards. The analysis confirmed the high levels of aurofusarin in fungal mycelia.

To determine whether aurofusarin synthesis is stimulated by predation in other *Fusarium* species, cultures of *F. poae*, *F. venenatum*, and *F. avenaceum* on solid media were subjected to grazing by the springtail *F. candida*, and *F. venenatum* and *F. sporotrichioides* were subjected to grazing by the woodlouse *Porcellio scaber* (Supplementary Fig. 4). Mycelia of *F. venenatum*, *F. sporotrichioides*, and *F. avenaceum* turned red in areas exposed to predation, indicating that the predation had stimulated aurofusarin synthesis.

**Aurofusarin deters a wide range of predators from feeding.** A key characteristic of defence metabolites is that they suppress predation. To test whether aurofusarin protected its producers from predation, *F. graminearum* accumulating aurofusarin and genetically engineered strains that were unable to produce aurofusarin were simultaneously offered to predators in food choice experiments (Fig. 3). Predators representing distant arthropod

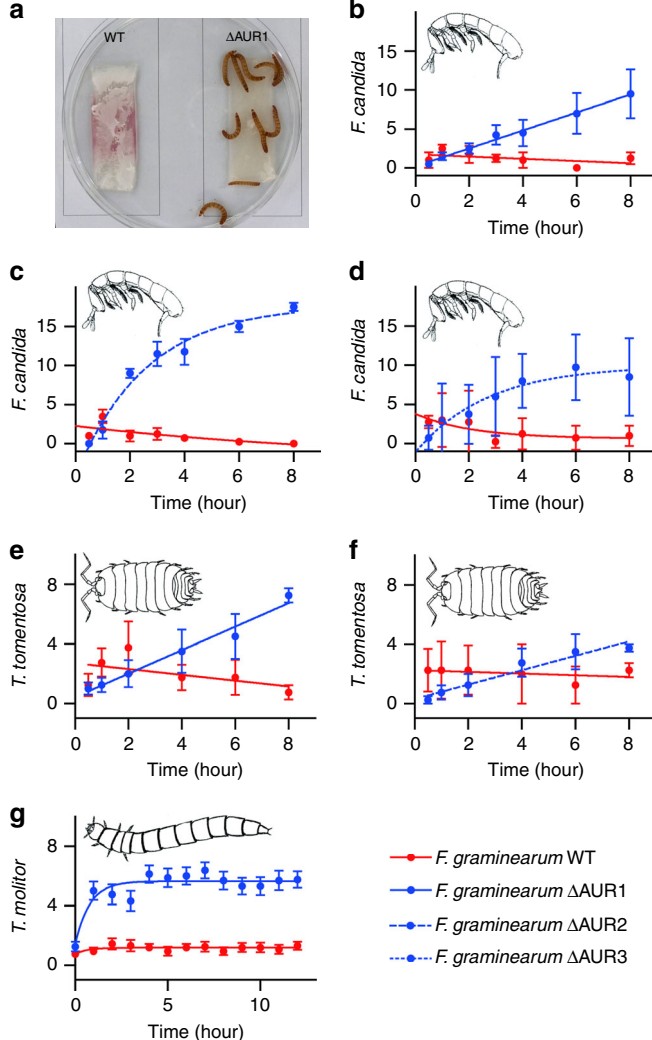

**Fig. 3** Predators avoid fungal cultures in which aurofusarin accumulates. **a** Mealworms in a Petri dish with cultures of *Fusarium graminearum* producing aurofusarin (WT) and a nonproducing mutant (ΔAUR1). **b–d** The food preferences of the springtail *Folsomia candida* for *F. graminearum* WT and aurofusarin-nonproducing mutants were studied by placing springtails that had been starved for 2 days into the centre of a Petri dish containing fungal cultures and by counting the animals feeding on each culture (20 animals per plate; four replicates). **e–f** The food preference of the isopod *Trichorhina tomentosa* was tested in the same manner with eight animals per arena and four replicates. **g** The food preference of the mealworm *Tenebrio molitor* was examined by placing larvae into Petri dishes containing fungal cultures on microscope slides, as shown in **a**. Sixteen replicates with 10 animals per plate were used. Error bars show 95% CI. Source data are provided in a Source Data file

lineages were used: the collembolan *F. candida* (primitive arthropod), the woodlouse *Trichorhina tomentosa* (crustacean), and the mealworm *Tenebrio molitor* (insect). Aurofusarin-producing and nonproducing cultures were placed onto opposite sides of Petri dishes, and the number of animals feeding on each culture was monitored. All predators displayed a strong preference for mutants that did not produce aurofusarin (Fig. 3). Within 1 hour, most mealworms had gathered on the cultures without aurofusarin, where they remained until the end of the experiment. The springtails and woodlice gradually gathered on cultures of non-producers; as shown in Fig. 3, after 8 h most animals were feeding on cultures without aurofusarin. The

disruption of biosynthetic pathways for the mycotoxins deoxynivalenol and zearalenone in *F. graminearum* had no effect on food preference (Supplementary Fig. 5), though both mycotoxins are toxic to insects[13,14]. The reversal of the springtails' food preference for *F. verticillioides* over *F. graminearum* via the disruption of aurofusarin synthesis in *F. graminearum* (Supplementary Fig. 5) indicates that aurofusarin had served as the major—or only—defence metabolite of *F. graminearum* deterring the springtails in this experiment.

The disruption of the biosynthetic pathway for aurofusarin synthesis may indirectly affect the synthesis of other metabolites[25,29] which may include unknown attractants. To clarify whether indirect effects of the disruption of the aurofusarin pathway may account for the arthropods' preference for fungi in which aurofusarin does not accumulate, mealworms were offered wheat flour amended with purified aurofusarin and unamended flour (Fig. 4). The larvae's strong preference for flour without aurofusarin revealed that aurofusarin possesses antifeedant activity and efficiently deters predators at a concentration similar to its concentration in fungal mycelia upon grazing (Fig. 2d, e). The exclusion of light in these experiments helped ensure that the animals not recognise aurofusarin by its colour.

**Aurofusarin is not toxic to arthropods**. Why do predators avoid aurofusarin-accumulating fungi? The avoidance of food containing toxins is an adaptation that reduces toxic exposure[7]. Aurofusarin has been reported to be toxic in poultry[30], but metabolites other than aurofusarin might have been responsible for the effects described in this work because the poultry feed used in these trials had not been amended with pure aurofusarin, but rather with a culture of a fungus known to be a potent producer of mycotoxins. To determine whether aurofusarin is toxic to insects, mealworms were fed wheat flour amended with aurofusarin for 10 d, and their weight gain was determined. Aurofusarin at concentrations of up to $1 \text{ mg g}^{-1}$ did not affect the mealworms' growth (Fig. 5a). This concentration is two to three orders of magnitude greater than concentrations at which mycotoxins display toxicity in insects[13,14,31]. At $10 \text{ mg g}^{-1}$, aurofusarin suppressed the mealworms' growth. At this concentration, aurofusarin substantially reduced feed intake (Fig. 4a), and we therefore assume that the suppression of mealworms' growth on flour with $10 \text{ mg g}^{-1}$ of aurofusarin was caused by reduced feeding rather than toxicity.

To test the developmental toxicity of aurofusarin on an arthropod that did not feed on filamentous fungi, we fed larvae of *Drosophila melanogaster* with food amended with aurofusarin at the same level as in the previous experiments as well as at a lower level of $2 \text{ mg g}^{-1}$ for 2 days. This feeding was followed by a transfer to a medium without aurofusarin to accomplish the development (Fig. 5b). No differences in the number of adults emerging from pupae in feeding trials with and without aurofusarin were found, which indicated that aurofusarin did not cause developmental toxicity in *D. melanogaster*.

The effect of aurofusarin on the growth and mortality of the springtail *F. candida* and the woodlice *Trichorhina tomentosa* was studied by monitoring the mortality of animals fed on *F. graminearum* and its aurofusarin-nonproducing mutant for 5 weeks, and the size of the animal bodies was estimated at the end of the experiment (Table 1). Forced feeding on *F. graminearum* cultures in which aurofusarin had accumulated did not cause any mortality in the springtail *F. candida* or the isopod *Trichorhina tomentosa*. The growth of animals fed on mycelia with aurofusarin was reduced as compared with aurofusarin-nonproducing mutants, but the effects were small, indicating that reduced feed consumption rather than toxicity was the cause. The lack of mortality during 5 weeks of feeding on

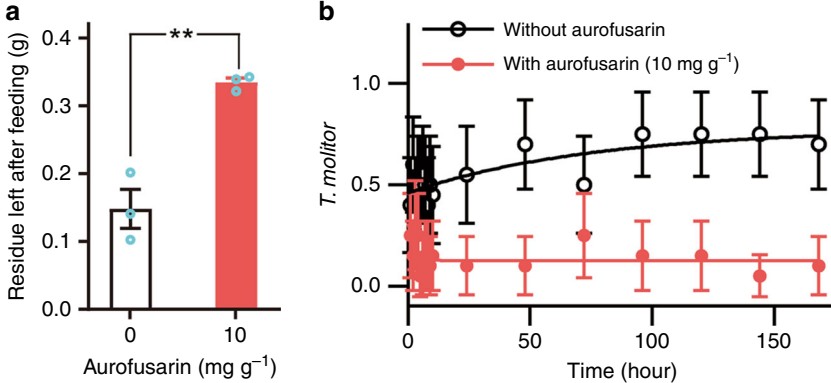

**Fig. 4** Aurofusarin in wheat flour repels mealworms. **a** The weight of the wheat flour and faeces left by five larvae of *T. molitor* after 4 d of feeding on 0.4 g of wheat flour with and without 10 mg g⁻¹ of aurofusarin. The significance of the difference was analysed with unpaired two-tailed *t* test ($n = 3$, $p = 0.0033$). **b** Single mealworms were placed on Petri dishes between two portions of 100 mg of wheat flour, one of which was amended with 10 mg g⁻¹ of aurofusarin. The arenas were kept in total darkness and opened only for a second in dim light to record the mealworm location ($n = 20$, error bars show CI 95%). Source data are provided in a Source Data file

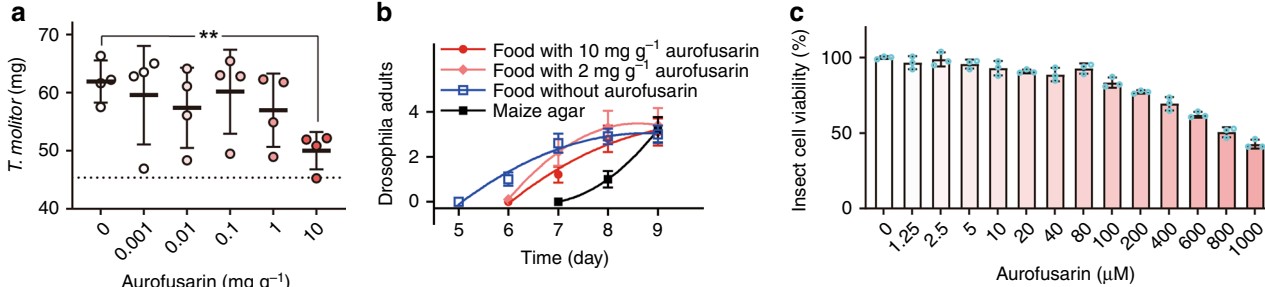

**Fig. 5** Toxicity of aurofusarin to arthropods. **a** The weight of *T. molitor* after 10 d of feeding on wheat with different concentrations of aurofusarin. The average initial weight is indicated by the dashed line. Means and SD of 4 replicates, each with 10 animals, are shown. The effect of aurofusarin at 10 mg g⁻¹ was analysed with two-sided *t* tests (10 individuals per experiment, $n = 4$, $p = 0.0027$). **b** The developmental toxicity of aurofusarin was tested by feeding larvae of *Drosophila melanogaster* on food with and without aurofusarin for two days, followed by incubation on standard food to accomplish the development (10 larvae per arena, 10 arenas per treatment, error bars show s.e.m.). Maize agar was used to simulate starvation. **c** The viability of Sf9 cells (fall armyworm *Spodoptera frugiperda*) after 24 h of incubation with aurofusarin ($n = 3$) shown as means with error bars showing s.e.m. The coloration of data points and bars indicates aurofusarin concentration in flour. Source data are provided in a Source Data file

| Table 1 Growth of predators fed aurofusarin mutants of *Fusarium graminearum* | | | | | | | | |
|---|---|---|---|---|---|---|---|---|
| **Predator** | ***F. g.*** | ***n*** | **Length (mm)** | **Rel. length** | ***p* value** | **Width (mm)** | **Rel. width** | ***p* value** |
| *F. candida* | WT | 56 | 0.89 ± 0.13 | 100% | – | 0.19 ± 0.03 | 100% | – |
| | ΔAUR1 | 21 | 1.08 ± 0.20 | 121% | < 0.0001 | 0.23 ± 0.06 | 123% | < 0.0001 |
| | ΔAUR2 | 35 | 1.05 ± 0.23 | 119% | < 0.0001 | 0.23 ± 0.06 | 122% | < 0.0001 |
| | ΔAUR3 | 35 | 0.94 ± 0.15 | 105% | 0.12 | 0.21 ± 0.04 | 112% | 0.0085 |
| *T. tomentosa* | WT | 10 | 1.50 ± 0.13 | 100% | – | 0.62 ± 0.07 | 100% | – |
| | ΔAUR1 | 12 | 1.81 ± 0.10 | 121% | < 0.0001 | 0.69 ± 0.09 | 112% | 0.048 |
| | ΔAUR2 | 13 | 1.76 ± 0.09 | 117% | < 0.0001 | 0.70 ± 0.05 | 114% | 0.0029 |
| | ΔAUR3 | 14 | 1.82 ± 0.15 | 122% | < 0.0001 | 0.73 ± 0.06 | 119% | < 0.0001 |

The animals were fed on *F. graminearum* for 5 weeks. *F. candida*, springtail *Folsomia candida*; *T. tomentosa*, isopod *Trichorhina tomentosa*; Rel. length, body length relative to animals fed on WT; Rel. width, body width relative to animals fed on WT. Size at the beginning of trial: *F. candida* length 0.48 ± 0.08 mm, width 0.11 ± 0.02 mm; *T. tomentosa* length 1.20 ± 0.08 mm, width 0.50 ± 0.05 mm. Length and width are shown as mean ± s.d. *p* values were determined using a two-tailed *t* test. Source data are provided in a Source Data file

aurofusarin-containing mycelia corroborates the lack of toxicity of aurofusarin in springtails and isopods.

The low toxicity of aurofusarin in mealworms, Drosophila, isopods, and springtails could be accounted for by inefficient absorption, detoxification in the digestive tract, or fast clearance. Toxicity assays with cell cultures circumvent these effects, and we therefore investigated the effect of aurofusarin on a cell culture of the fall armyworm, *Spodoptera frugiperda*, which is an established

toxicity model for insects[14]. Aurofusarin also exhibited relatively low toxicity in insect cells (Fig. 5c). The low toxicity of aurofusarin contradicts the hypothesis that fungal defence metabolites are toxic to predators[12–15,19,20].

**Mechanism of the induction of aurofusarin synthesis by grazing.** Increased levels of aurofusarin in shaken cultures

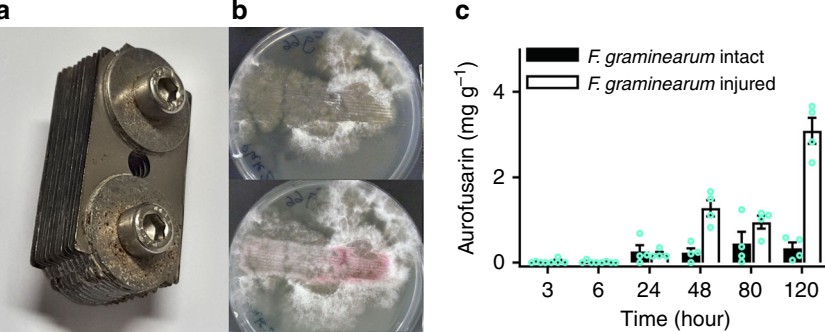

**Fig. 6** Mechanical injury induces aurofusarin synthesis. **a** An array of 10 razor blades assembled at distances of 1 mm. **b** *Fusarium graminearum* IFA66 on GM7 medium at 22 °C immediately after injury with the razor blade array (upper photo) and 24 h later (lower photo). **c** *F. graminearum* cultures were grown on rice media for 5 days, injured with the blade array in the same manner as in **b**, and harvested at the indicated times. The aurofusarin content was determined via HPLC-MS/MS ($n = 4$, error bars show s.e.m). Source data are provided in a Source Data file

(Fig. 2b) and in mycelia that had been exposed to a wide range of predators with different feeding modes (Fig. 2d, e, i–j) indicate that mechanical damage alone stimulates the synthesis of aurofusarin in *F. graminearum*. To test this hypothesis, we injured the mycelium of *F. graminearum* using an array of razor blades and monitored the aurofusarin content in the mycelia (Fig. 6). The results confirmed that mechanical damage was sufficient to induce aurofusarin synthesis in *F. graminearum* and showed that the effect was local (remaining confined to damaged parts of mycelia) and that the accumulation of increased levels of aurofusarin continued for at least 120 h after the injury. The mycelia of *F. graminearum* and *F. culmorum* in shaken liquid cultures accumulated more aurofusarin than still cultures, indicating that shaking caused mechanical injury (Supplementary Fig. 3). *F. avenaceum* and *F. poae* accumulated low amounts of aurofusarin in both culture types, but *F. venenatum* produced higher amounts of aurofusarin in still cultures than in shaken cultures, contradicting the results obtained with *F. graminearum* and *F. culmorum*. To clarify the discrepancy, the mycelium of *F. venenatum* that had been growing on an agar medium was injured with a razor blade array (Supplementary Fig. 6). Within 24 h, the injured mycelia turned red, showing that mechanical damage induced aurofusarin synthesis also in *F. venenatum*. As shaking has not stimulated aurofusarin synthesis in *F. venenatum* (Supplementary Fig. 3), it apparently has not caused damage in this fungus comparable to cutting (Supplementary Fig. 6) or predation (Supplementary Fig. 4b).

**Further fungal bis-naphthopyrones act as antifeedants**. The magnitude of the deterrence effect of aurofusarin and the wide range of predators responsive to the antifeedant indicated that aurofusarin is *F. graminereaum*'s major defence metabolite. Dimeric naphthopyrones similar to aurofusarin are produced by many genera of filamentous ascomycetes. Core structures of over 50 such metabolites are shown in Fig. 7. Their biological function is unknown. Viomellein and xanthomegnin—which are produced by many species of *Aspergillus*, *Penicillium*, *Trichophyton*, and other genera—were selected to investigate their inducibility by predation and their antifeedant activity towards arthropods. Induction of the viomellein synthesis by grazing was tested by subjecting cultures of the viomellein producer *Aspergillus ochraceus* to grazing by the springtail *F. candida*. The analysis of extracts of grazed and controlled fungal cultures by HPLC showed that grazing stimulated the synthesis of viomellein in fungal mycelia (Fig. 8a).

The deterrent effect of xanthomegnin and viomellein on the springtail *F. candida* was tested by offering baker's yeast spiked with these metabolites to the springtails in food choice experiments. At a spiking level of 10 mg g$^{-1}$, the springtails strongly avoided yeast containing either xanthomegnin or viomellein (Fig. 8b, d). After 20 min, hardly any animal was found feeding on food spiked with xanthomegnin or viomellein. The deterrence was less prominent yet still highly significant at a spiking level of 2 mg g$^{-1}$ (Fig. 8c, e). The results showed that the fungal bis-naphthopyrones viomellein and xanthomegnin are antifeedants that exert effects similar to aurofusarin on the springtail *F. candida*.

## Discussion

Secondary metabolite synthesis in fungi is highly diverse, and most secondary metabolite pathways are species-specific[32,33]. Identical or similar structures are rarely found among secondary metabolites produced by more than two fungal genera. In contrast to other secondary metabolites, dimeric naphthopyrones have been found in all genera of filamentous ascomycetes investigated thus far, suggesting that they fulfil a common and widespread biological function. Our study indicates that this function is defence against animal predators.

Fungal strains defective in secondary metabolism owing to dysfunctional global regulator velvet complex[20,34], which controls secondary metabolism and development[35], and strains with constitutively stimulated secondary metabolism[36] were used in food choice experiments. Predators preferred strains impaired in secondary metabolite synthesis and avoided strains with constitutively stimulated secondary metabolite synthesis, but the pleiotropic character of these mutations prevented identification of metabolites responsible for the effects. Mycotoxin sterigmatocystin was most often implicated in defence, but pathway-specific mutants failed to confirm its role[20]. Our results indicate that the metabolites responsible for the loss of protection against predation in fungal strains with globally suppressed secondary metabolism were dimeric naphthopyrones.

If bis-naphthopyrones are defence metabolites ubiquitous among ascomycetes, why was the induction of their synthesis by predation not observed earlier? Transcriptomic studies in fungi have included various sorts of treatments, but surprisingly the effect of predation on fungal transcriptome has not been investigated. The effect of predation on fungal metabolome was addressed in a single study, in which *Aspergillus nidulans* was exposed to grazing by the springtail *F. candida*[15]. The metabolites found to be stimulated by predation did not include naphthopyrones. *A. nidulans* produces naphthopyrone YWA1, which is similar to the aurofusarin precursor rubrofusarin[28] and is dimerised into green pigment[37] in the same way that rubrofusarin is dimerised into aurofusarin[28]. Why was the synthesis of the pigment not stimulated by grazing? The likely

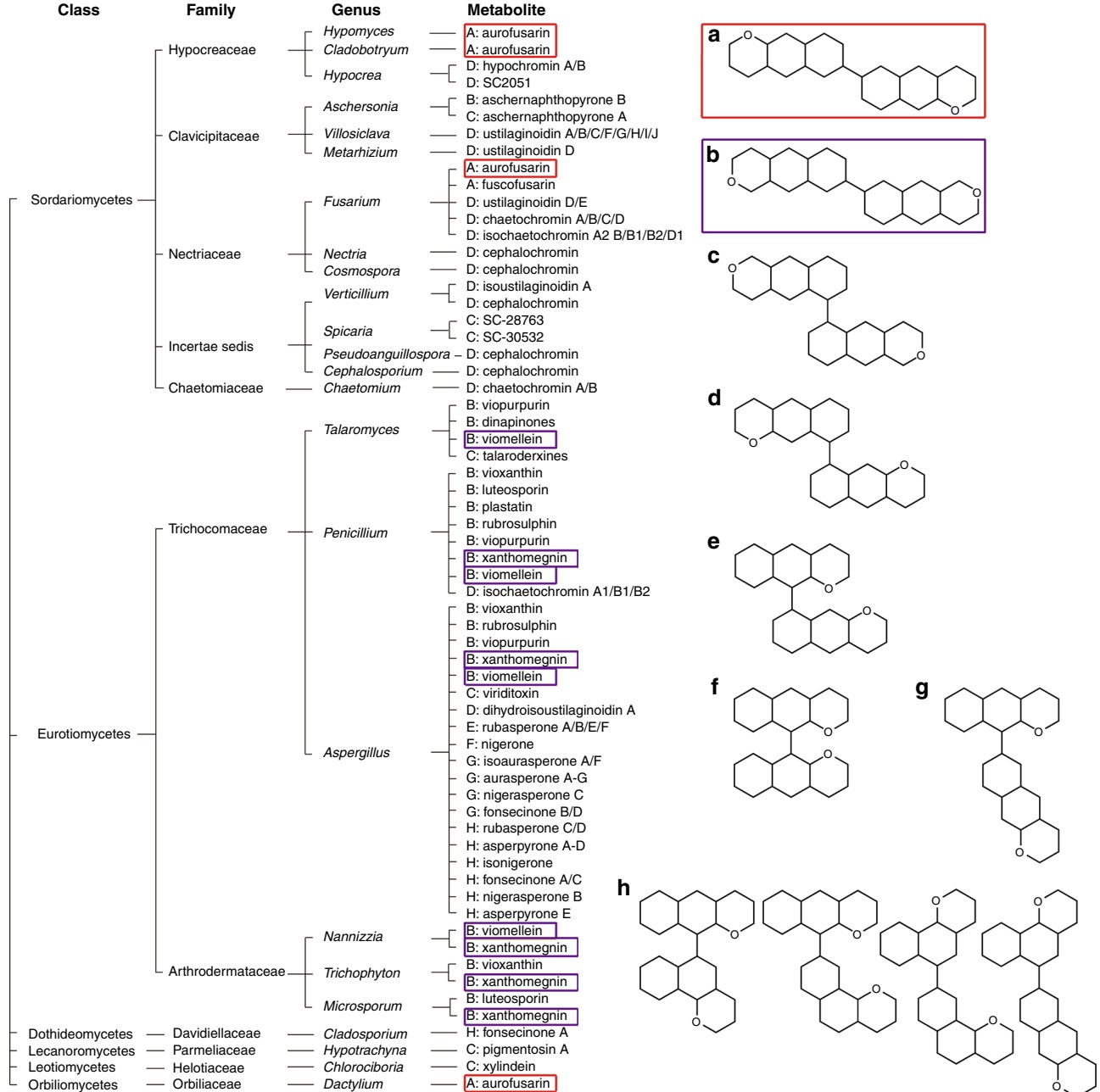

**Fig. 7** Production of dimeric naphthopyrones by ascomycetous fungi. The taxonomic affiliation of selected fungal genera that produce dimeric naphthopyrones[27] is shown on the left with schematic structures of bis-naphthopyrones on the right. The Structure Classes A to G contain dimeric naphtho-α-pyrones and naphtho-γ-pyrones consisting of linear heptaketides; the classes differ by the position of the link between monomers and by the presence of either α- or γ-pyrones. The metabolites of Class H contain angular heptaketides, and all metabolites of this class listed here contain γ-pyrones. The metabolites used in this study have been labelled

reason was that as in most studies of interactions between *A. nidulans* and arthropods conducted thus far, in this study, an *A. nidulans* strain carrying mutation *ve*A1 was used, which is defective in secondary metabolite production and defence responses[38]. VeA is part of the velvet complex, which is needed for aurofusarin synthesis in *F. graminearum*[39]. The metabolites induced by grazing in *A. nidulans ve*A1 may belong to a second-tier level of defence, though our results indicate that in *F. graminearum*, bis-naphthopyrones are mainly—if not only—defence metabolites that target predators.

One *Fusarium* species that does not produce dimeric naphthopyrones is *F. proliferatum*. The absence of antifeedant metabolites in this species is in line with the observation that wheat kernels colonised with *F. proliferatum* attracted rather than repelled the mealworm *Tenebrio molitor*[40]. *F. proliferatum* survives a passage through the digestive system of *T. molitor*, and its propagules continue to be disseminated by faeces of the beetle long after ingestion[41], suggesting that the loss of bis-naphthopyrone synthesis may have been selected in this species during its adaptation to dissemination by insects.

Aurofusarin synthesis in *Fusarium* species was induced by predators with different feeding modes and by cutting mycelia with razor blades, indicating that mechanical damage was sufficient to trigger chemical defence against predation. The potential

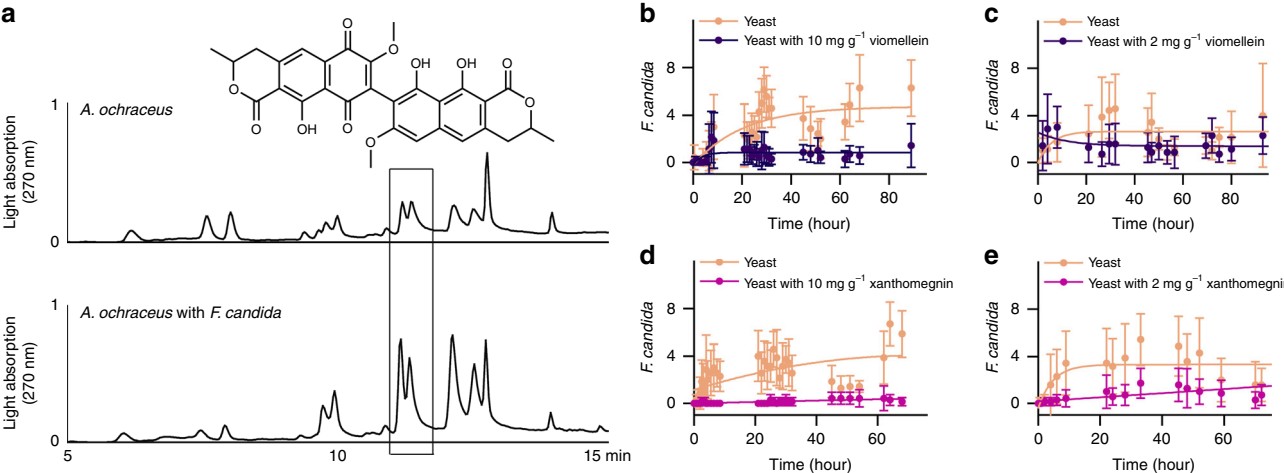

**Fig. 8 Antifeedant activity of viomellein and xanthomegnin. a** *Aspergillus ochraceus* was grown on rice kernels for 5 d. The springtail *Folsomia candida* was added to fungal cultures and allowed to feed for 6 d. The cultures were extracted with chloroform/methanol (80:20) and analysed via HPLC with light absorption detection. Viomellein appeared as two peaks originating from stereoisomers (atropisomers) and separated owing to restricted intramolecular rotation around the biaryl axis. **b–e** The food preference of springtails *Folsomia candida* for yeast amended with viomellein or xanthomegnin. The animals were placed into the middle of arenas containing dry yeast on one side and dry yeast amended with viomellein or xanthomegnin on the other side. The number of springtails visiting each diet was recorded in a time series. Seven replicates were used, each with 20 animals. No statistical test was used for the data obtained for $10\,\text{mg}\,\text{g}^{-1}$ of bis-naphthopyrones **b**, **d**. For the data obtained with $2\,\text{mg}\,\text{g}^{-1}$ of bis-naphthopyrones **c**, **e**, cumulative values for all time points beginning with 20 h were compared using two-tailed $t$ tests. The preference for yeast without naphthopyrones was highly significant for both metabolites (viomellein: $n = 14$, $p = 0.0054$; xanthomegnin: $n = 10$, $p < 0.0001$). Error bars show 95% CI. Source data are provided in a Source Data file

for the activation of defence responses by shaking or stirring fungal cultures in buffled flasks and fermenters should be considered in physiological studies on fungi in liquid media. The induction of bis-naphthopyrones synthesis in fungi by wounding contrasts with the chemical defence of land plants against herbivory, which requires specific signals in addition to wounding[42]. Defence response induction in plants by wounding alone would lead to frequent false alarms because plant shoots are often injured by being hit by solid objects in the wind and owing to passing animals, whereas fungal hyphae are protected by the solid substrates inside of which they grow. The second difference between the chemical defence of plants and fungi against predation is that plant defence against herbivory spreads systemically and even reaches neighbouring plants via volatile signals[43], whereas fungal defence remains confined to the area affected by the damage (Fig. 6, Supplementary Figs. 4 and 6).

The presence of defence metabolites of the same structural class that presumably possess the same mode of action in many fungal species is likely to exert a strong selection pressure on the predators. If insects are notorious for their rapid development of resistance to insecticides[44] and mushroom-feeding insects tolerate mushroom toxins[18], why are bis-naphthopyrones still active against a wide range of fungivores that have likely been exposed to these metabolites for hundreds of millions of years? Aurofusarin content in *F. graminearum* that has been exposed to predation is very high (Figs. 2d, e). The maintenance of the high production of aurofusarin that may infer substantial fitness costs —as evidenced by markedly increased growth rates of mutants with disrupted aurofusarin synthesis[25,29]—must have been subjected to a strong selection pressure. We hypothesise that high levels of aurofusarin in fungal mycelia prevented predators from adaptation by saturating molecular targets of aurofusarin with binding affinities reduced by mutations, or by overwhelming enzymatic degradation. Mycotoxins never accumulate in comparably high concentrations, which would probably cause self-poisoning that even protection mechanisms of mycotoxin producers[45] could not prevent. The synthesis of an antifeedant of low toxicity in large amounts exemplifies a new concept in fungal

chemical defence, with aurofusarin, viomellein, and xanthomegnin serving as the first examples. The ubiquity of bis-naphthopyrone pigments in ascomycetes indicates that this defence mechanism is widespread. The defence function of aurofusarin extends and modifies Janzen's 40-year-old hypothesis that fungi protect their substrates from animals via poisonous chemicals[11]. Rather than structurally diverse toxins produced in low amounts, structurally similar antifeedants accumulating at high amounts protect fungi from soil-dwelling predators. The intriguing question for future research is how a single metabolite class deters a wide range of predators. The presence of gustatory receptors triggered by bis-naphthopyrones in phylogenetically distant arthropods, including crustaceans, springtails, and insects, indicates that natural ligands of these receptors are compounds common in food substrates of all arthropods, such as proteins or polysaccharides.

## Methods

**Animals**. The larvae of the beetle *Tenebrio molitor* (Insecta: Coleoptera) and the isopod *Trichorhina tomentosa* (Crustacea: Oniscidea) were purchased from Zoo & Co. Zoo-Busch and b.t.b.e. Insektenzucht GmbH (Schnürpflingen, Germany). The culture of *F. candida* (strain: Berlin) was obtained from the Institute of Zoology, University of Goettingen, Germany, and was kept on Petri dishes filled with a layer of gypsum plaster with charcoal (9: 1). The culture of *Porcellio scaber* was initiated using animals collected in Göttingen, Germany, in spring 2017. The culture of *Drosophila melanogaster* was initiated with animals caught in 2006 in Kiel, Germany. Cultures of the nematodes *Aphelenchoides saprophilus* and *Bursaphelenchus mucronatus* were obtained from Prof. Liliane Ruess, Humboldt University of Berlin, Germany, and Professor Jiafu Hu, Zhejiang Agriculture and Forestry University, China, respectively.

**Fungal strains**. The *F. graminearum* strain IFA66[46] (DON chemotype) was obtained from Dr Marc Lemmens (BOKU, Tull, Austria) via Professor Thomas Miedaner (University of Hohenheim, Germany). Aurofusarin-deficient mutants were generated by disrupting the polyketide synthase gene PKS12[25] in *F.graminearum* 1003 and were labelled △AUR1, △AUR2, and △AUR3. The zearalenone-deficient mutant △ZEN was generated from the same parent by disrupting the polyketide synthase gene PKS4 involved in zearalenone synthesis. The deoxynivalenol-deficient mutant △DON was generated from the strain *F. graminearum* 3211 by disrupting Tri5 gene[47]. *F. culmorum* 3.37[40] was a gift from late Professor Heinz-Wilhelm Dehne (University of Bonn, Germany). *A. ochraceus* 6692 was obtained from Professor Rolf Geisen, Max Rubner-Institut, Karlsruhe,

Germany. *F. avenaceum* BBA92013 was obtained from Professor Tapani Yli-Mattila, Turku, Finland. *F. tricinctum* RD30, *F. venenatum* RD15, and *F. venenatum* RD90 were isolated from weed samples in Germany and provided by Dr Raana Dastjerdi, University of Goettingen, Germany. *F. poae* DSMZ62376 was obtained from DSMZ (Braunschweig, Germany), and *F. verticillioides* M-8114 was obtained from the Fusarium Research Center (University Park, PA, USA). *F. sporotrichioides* IPP0249 was obtained from Prof. A. von Tiedemann (University of Goettingen, Göttingen, Germany).

**Fungal media and cultures.** The potato broth medium (PDB) was prepared by boiling 200 g of potatoes with the peel in 1 L of tap water for 20 min and autoclaving the filtrate with 20 g of glucose. GM7 medium[48] contained 3 g L-asparagine, 1 g $KH_2PO_4$, 0.5 g $MgSO_4·7H_2O$, 20 g glucose, 50 mg $CaCl_2$, 10 mg $FeCl_3$, 1 mg thiamine and 20 g agar in 1 L; pH adjusted to 5.6. Liquid cultures of *Fusarium* spp. for the analysis of aurofusarin content were grown in 30 ml of PDB in 100-ml Erlenmeyer flasks at 23 °C. Rice medium for the analysis of aurofusarin accumulation in *Fusarium* spp. cultures after injury with a razor blade array was prepared by autoclaving a mixture of 1.5 ml of tap water with 0.5 g of rice powder (Alnatura GmbH, Bickenbach, Germany) in glass Petri dishes with a diameter of 5 cm. Rice medium for the investigation of the effect of predation on pigment accumulation was prepared by autoclaving 20 g of rice powder and 20 g of agar in 1 L of tap water and was poured into plastic Petri dishes with a diameter of 9 cm.

**Purification of aurofusarin.** Aurofusarin was extracted from *F. graminearum* IFA66 that had been grown in PDB for 2 weeks at 25 °C with shaking at 200 rpm. Fungal mycelium was freeze–dried, ground, and extracted with 50 ml of chloroform/methanol (80:20) per gram of mycelium. The extract was cleared by centrifugation and the solvent removed in vacuum. Aurofusarin was purified by ethanol precipitation from phenol[49] at 50 °C followed by crystallisation from glacial acetic acid[50]. The purity of crystallised aurofusarin was verified by HPLC-ELSD (see "Aurofusarin analysis by HPLC-MS/MS and HPLC-ELSD").

**Exposure of *F. graminearum* and *F. culmorum* to predation.** For transcriptome analysis, 5000 fungal spores in 5 μL of water were inoculated onto three rice kernels that had been autoclaved with 200 μl of demineralised water (for arthropods) or onto 40 mg of rice flour that had been autoclaved with 150 μl of demineralised water (for nematodes) in 15-ml Falcon tubes and incubated at 15 °C in the dark. After 7 d, the following predators were added to the fungal cultures: 20 mg (~200 individuals) of the springtail *F. candida* starved for 2 days or 2000–3000 individuals of the nematode *Aphelenchoides saprophilus* or 1000–2000 individuals of the nematode *Bursaphelenchus mucronatus*. Controls were incubated under the same conditions without animals. Each group consisted of four replicates.

**Transcriptome analysis by RNAseq.** *F. graminearum* IFA66 was exposed to predation by *F. candida*, as described above. After 48 h, four cultures with predators and four control cultures were harvested, the animals were removed, and the total RNA was extracted using the RNAsnap method[51], which was modified as follows: Fungal cultures were suspended in 400 μl of RNA extraction solution (95% deionised formamide, 18 mM EDTA, 0.025% SDS, and 1% 2-mercaptoethanol) and disrupted via shaking with zirconia beads (2.0 mm in diameter, Carl Roth, Karlsruhe, Germany) in the reciprocal mill MM 200 (Retsch, Haan, Germany) for 2 min at maximum power, followed by incubation at 95 °C for 7 min. Cell debris was removed by centrifugation at $16,000 \times g$ for 5 min at room temperature. The supernatant was transferred into an RNA precipitation mixture consisting of 800 μl of isobutanol, 400 μl of 5 M guanidine thiocyanate, and 5 μl of linear polyacrylamide used as a co-precipitant (Co-Precipitant Pink, Bioline, London, UK). The mixture was centrifuged at $16,000 \times g$ for 5 min at room temperature, and the pellet was washed with 75% ethanol, dried, and dissolved in RNAase-free water.

Strand-specific cDNA libraries were prepared using Illumina's TruSeq stranded mRNA kit (75 bp paired-end) and sequenced on Illumina NextSeq 500V2. Data were analysed using the public server of Galaxy[52] (https://usegalaxy.org/). Before the analysis, the reads were trimmed to remove low-quality sequences. The reads were mapped to the reference genome (*F. graminearum* PH-1) using Hisat2 v2.0.5.1 with the default options for single-end data[53]. Cufflinks v2.2.1[54] was used to determine the abundance of transcripts in FPKM (Fragments Per Kilobase of exon per Million fragments mapped): The maximum intron length was set to 1000 nt, and the last annotated genome (ASM24013v3) was used as a reference. Cuffdiff v2.2.1.3[54] was used to determine the changes in gene expression compared with the control using an FDR (false discovery rate) of 0.05. Only genes with a $\log_2$ FPKM (fold change) higher than 1.0 and a $q$ value lower than 0.01 were considered to have been significantly induced.

To identify upregulated secondary metabolite gene clusters, all upregulated genes were checked in the National Center for Biotechnology Information (NCBI) and in the European Bioinformatics Institute (EMBL–EBI) (UniProt) databases for signatures of polyketide synthetases (PKS), nonribosomal peptide synthetases, and terpenoid synthetases. Once more than 50% of the genes in a cluster and/or the signature enzyme of a cluster had been significantly induced (log2 FPKM (fold change) higher than 1.0 and $q$ value lower than 0.01), the literature related to the cluster was consulted to reveal the associated secondary metabolites. In this manner, gene clusters for the biosynthesis of aurofusarin[55], C62[56,57], fusarin C[56,58,59], fusaristatin A[56,57], and terpenoids[56] were identified. The results were corroborated with the help of AntiSMASH 3.0[60]. The upregulated genes that had been located immediately before and after the signature enzymes were considered part of the cluster. Putative clusters without known signature enzymes were identified using AntiSMASH 3.0[60] and corroborated manually.

**Transcription analysis of aurofusarin pathway by RT qPCR.** After exposure to *F. candida* grazing for 0, 1, 2, 4, 8, 24, 48, 80, and 120 h (*F. graminearum*) or 80 and 120 h (*F. culmorum*) and after nematode feeding for 24 and 80 h (*F. graminearum*), fungal mycelia were harvested, frozen in liquid nitrogen, and ground. RNA was extracted using the guanidinium thiocyanate–phenol–chloroform method[61], precipitated with 4 M LiCl for 3 h on ice, and reverse transcribed with RevertAid Reverse Transcriptase (Thermo Fisher Scientific, California, USA) and random primers according to the manufacturer's instructions using 400 ng of RNA in 20-μl reactions. The cDNA obtained was used as a template for PCR, which contained ThermoPol Reaction Buffer (20 mM Tris-HCl, 10 mM $(NH_4)_2SO_4$, 10 mM KCl, 0.1% Triton-X-100, pH 8.8 at 25 °C) with 2.5–4.5 mM $MgCl_2$, 200 μM dNTP, 0.3 μM forward and reverse primers (Supplementary Data 2), SYBR Green I (Invitrogen, Karlsruhe, Germany), 1 mg ml$^{-1}$ bovine serum albumin, 0.03 U μl$^{-1}$ Taq polymerase (New England Biolabs, UK), and 1 μl cDNA as a template. PCR conditions were as follows: 95 °C for 2 min, 35 cycles of 94 °C for 20–30 s, 59 °C for 30–40 s and 68 °C for 30 s, with a final extension of 68 °C for 15 min followed by a melting curve analysis beginning at 95 °C with a decrement of 0.5–55 °C. GAPDH- (glyceraldehyde-3-phosphate dehydrogenase) and EFIA (elongation factor 1-alpha) genes were used as an internal reference, and the primers are shown in Supplementary Data 2. Three to eight biological replicates were analysed (Supplementary Data 3). The amplification efficiency for each gene obtained with the help of serial dilutions was used to calculate relative transcript levels (fold change)[62]. The significance of differences between cultures subjected to predators and controls was determined, as described in "Statistics and reproducibility".

**Analysis of food preference.** The preference of the springtail *F. candida* and the isopod *Trichorhina tomentosa* for fungal strains was studied on Petri dishes (92 mm in diameter) filled with a mixture of gypsum plaster and charcoal (9:1). Fungal cultures were grown on potato agar prepared from potato broth (see above) solidified with 15 g L$^{-1}$ agar and kept at 25 °C for 7–8 d in the dark. Agar plaques were cut from the edge of fungal colonies using a sterile cord borer (12 mm in diameter) and placed onto discs of matching size cut from Parafilm and placed on the opposite sides of the Petri dishes. Twenty *F. candida* individuals or eight *Trichorhina tomentosa* individuals that had been starved for 2 days were placed into the centre of the Petri dishes, and the plates were incubated at 15 °C in the dark. The number of collembolans and isopods on each mycelium was recorded.

Food choice experiments with the mealworm *T. molitor* were carried out with fungal mycelia as well as with wheat flour amended with aurofusarin. In an experiment with fungal mycelia, *F. graminearum* strains were cultured on glass slides covered with PDA. After 7–8 d at 25 °C, the slides were placed on the opposite sides of Petri dishes (150 mm in diameter), and eight *T. molitor* larvae were placed in the middle of the plates. The number of animals visiting each fungal culture was recorded by taking photos at time intervals for 12 h, and animals inside 13 × 6-cm rectangles that had been drawn around each slide were counted. The experiments with wheat flour were carried out with groups of animals and with single animals. Wheat flour (summer wheat variety Taifun) was amended with aurofusarin that had been dissolved in chloroform to reach a final concentration of 10 mg g$^{-1}$. After shaking at room temperature for 1 h, the chloroform was removed in a vacuum and the flour was left in an open Petri dish overnight in a fume hood. Flour samples were treated with pure chloroform in the same manner. Portions of 0.4 g of flour with and without aurofusarin were placed on the opposite sides of Petri dishes. Five larvae of *T. molitor* were added to each of three Petri dishes and allowed to feed for 4 d at room temperature and in ambient light. The animals were subsequently removed, and the weight of the remaining wheat flour with and without aurofusarin (mixed with faecal pellets left by the animals) was determined. The experiments with single animals were carried out with portions of 100 mg of flour in Petri dishes covered with light-tight plant pot saucer trays. Single 3-month-old mealworms (23.9 ± 1.8 mm) were placed into the centre of 20 Petri dishes with one portion of flour with aurofusarin and another portion without aurofusarin. The light-blocking trays were opened for two seconds in dim light at fixed time intervals to record the mealworm location ($n = 20$).

Food choice experiments addressing the effect of xanthomegnin and viomellein on *F. candida* were carried out with dry yeast (Dr. August Oetker Nahrungsmittel KG, Bielefeld, Germany). The yeast was amended with aurofusarin dissolved in chloroform to achieve final concentrations of 2 mg g$^{-1}$ and 10 mg g$^{-1}$, chloroform was removed in vacuum and the yeast was left in open Petri dish overnight in a fume hood. Yeast for the controls was treated with pure chloroform. Twenty collembolans starved for 2 days were added to Petri dishes filled with charcoal/plaster (1:9) and that contained 3 mg of dry yeast on one side and 3 mg of dry yeast amended with viomellein or xanthomegnin on the other side and the number of springtails visiting each diet was recorded in a time series.

**Toxicity of aurofusarin to mealworm Tenebrio molitor**. Wheat flour was amended with aurofusarin dissolved in chloroform, and the solvent was evaporated, leading to aurofusarin concentrations of 0, 1, 10, 100, 1000, and 10,000 µg g$^{-1}$. Groups of ten larvae of *T. molitor* were weighed individually and placed in Petri dishes containing 1 g of wheat flour amended with different amounts of aurofusarin. After 10 d at 18 °C in the dark, the weight of the animals was determined again. Each treatment consisted of four replicates, and two independent experiments were carried out.

**Toxicity of aurofusarin to insect cells in tissue culture**. The *S. frugiperda* 9 (Sf9) cell line was maintained in a Sf-900 II medium (Thermo Fisher Scientific China, Shanghai, China) and grown at 28 °C. The cells were seeded into 96-well cell culture plates with $8 \times 10^3$ cells in 100 µl medium per well, incubated for 24 h and treated with 100 µl medium containing aurofusarin dissolved in dimethyl sulfoxide (DMSO) for 24 h with the final DMSO concentration not exceeding 0.1%. MTT (3-(4,5-dimethylthiazol-2-yl)−2,5-diphenyltetrazolium bromide) solution (0.5 mg mL$^{-1}$) was added to each well, and incubation was continued for 4 h. Formazan precipitate was dissolved in 150 µL of DMSO, and the absorbance of the solution at 490 nm was determined. Medium containing 0.1% DMSO was used as a negative control. The test was performed in triplicate.

**Developmental toxicity of aurofusarin in Drosophila**. Flies were reared at room temperature in BugDorm cages (MegaView Science, Taichung, Taiwan). Eggs were collected in Petri dishes left in the cages and incubated at 22 °C in the dark for hatching. Larvae were transferred into 2-ml-Eppendorf tubes containing 160 mg of medium, which were closed with sponge stoppers and incubated in a humid chamber in the dark at 25 °C. The standard food medium consisted of 62.5 g of baker's yeast, 62.5 g of cornmeal, 62.5 g of sucrose, and 12.5 g of agar per litre; the maize agar consisted of 188 g of cornmeal and 12.5 g agar per litre. The medium amended with aurofusarin was prepared by adding a chloroform solution of aurofusarin to a mixture of yeast, cornmeal, and sucrose, followed by removing the chloroform in a vacuum and adding the required amount of 1.25% agar in water autoclaved and cooled to 60 °C. After incubation on the medium with aurofusarin for 2 days (with the full medium and the cornmeal medium used as controls), all larvae were transferred to a fresh medium to accomplish the development, and the flies that emerged from pupae were counted daily.

**Inducibility of aurofusarin synthesis by mechanical injury**. To visualise aurofusarin induction by injury, fungal spores were spread onto GM7 medium incubated at 22 °C. The developed mycelia were injured with an array of 10 razor blades spaced at 1 mm apart. 24 h later, the aurofusarin accumulation was monitored as red pigment. For HPLC analysis of the aurofusarin accumulation after injury, 10,000 spores of *F. graminearum* IFA66 in 10 µl of water were inoculated into the middle of rice medium in Petri dishes and incubated at 23 °C in the dark for 5 d. Mycelium on the plate surface was injured by five cuts with an array of 10 razor blades spaced at 1 mm apart. The cultures were kept at 23 °C, harvested at a time series, and extracted for aurofusarin analysis.

**Exposure of Aspergillus ochraceus to predation**. For treatment with the springtails, *A. ochraceus* 6692 was grown on three autoclaved rice kernels, as described for *F. graminearum* under "Exposure of *F. graminearum* and *F. culmorum*…". After 5 d at room temperature in the dark, the springtail *F. candida* was added (20 mg, ca. 200 individuals) and allowed to feed for 6 d. For treatments with mealworms, *A. ochraceus* was grown in a medium made by autoclaving 1 g of maize flour with 4 ml of tap water in 100-ml Erlenmeyer flasks. After 5 d at room temperature in the dark, eight larvae of *T. molitor* were added to each flask and allowed to feed for 6 d. The animals were removed, the cultures were freeze–dried, and viomellein was extracted with chloroform/methanol (80:20) using a 2-ml solvent for rice cultures and a 15-ml solvent for maize cultures. The extracts were cleared by centrifugation and the solvent was removed in a vacuum. The residue was dissolved in DMSO.

**Determination of aurofusarin and viomellein content by HPLC-DAD**. Aurofusarin was extracted from freeze–dried cultures using chloroform/methanol (80:20). The extracts were cleared by centrifugation, the solvent was removed in a vacuum, and the residue was dissolved in DMSO. The aurofusarin content was determined by HPLC with a diode-array detector (DAD, Varian Prostar) using a polar-modified C18 column (Polaris Ether, 100 × 2.0 mm; Varian, Darmstadt, Germany) that was kept at 40 °C, which was eluted with a gradient of Solvents A (water with 0.05% acetic acid and 5% acetonitrile) and B (methanol with 0.05% acetic acid): 0.1 min 60% B, 11.9 min 60–98% B, 2 min 98% B, 1 min 98–60% B, 8 min 60% B. Light absorption was monitored at 243 nm. The aurofusarin standard turned out to be extremely unstable in protic solvents and under light, as has been observed by other researchers[63]. Stock solutions in DMSO were therefore kept at − 80 °C. We found that it was crucial to prepare calibration standards in pure methanol instead of using the mobile phase to prevent degradation during the run. Viomellein was extracted in the same manner as aurofusarin. The residue after solvent removal was dissolved in DMSO and analysed by HPLC on a C18 column (Kinetex 2.6 µm, 50 × 2.1 mm, Phenomenex,

Aschaffenburg, Germany) that was kept at 40 °C, which was eluted with a gradient of Solvents A (water with 3% acetic acid and 5% acetonitrile) and B (methanol with 3% acetic acid) as follows: 2 min 40% B, 12 min 40–98% B, 2 min 98–98%, and 1 min 98–40% B, 8 min 40% B. Viomellein was detected by light absorption at 270 nm. Aurofusarin and viomellein standards were purchased from Bioviotica (Göttingen, Germany).

**Aurofusarin analysis by HPLC-MS/MS and HPLC-ELSD**. Because the concentrations of aurofusarin in fungal mycelia that were determined by HPLC-DAD turned out to be unprecedentedly high, two other methods were used for confirmation. The first method was HPLC-MS/MS with separation on a polar-modified C18 column (Polaris Ether, 100 × 2.0 mm; Varian, Darmstadt, Germany) that was kept at 40 °C, which was eluted with a gradient of Solvents A (water with 0.1% formic acid) and B (methanol) as follows: 0.2 min 40% B, 6.8 min 40–98% B, 2 min 98% B, 1 min 98–40% B, 5 min 40% B. The eluent was ionised by electrospray in a positive mode with a capillary voltage of 4000 V. The triple quadrupole 6460 (Agilent, Darmstadt, Germany) that was used as a detector was operated in a selected reaction monitoring mode with the transition m/z 571.3- > 556.0. Aurofusarin standards were prepared shortly before analysis by diluting a stock solution in DMSO with pure methanol (see previous paragraph). As an additional verification method independent of the aurofusarin standard, HPLC coupled with an evaporative light scattering detector (ELSD) was used. The analyte was chromatographically separated (as described above), and the eluent was directed into the ELSD 1260 detector (Agilent, Darmstadt, Germany) operated at a nebuliser temperature of 40 °C, an evaporator temperature of 42 °C, and an evaporator gas flow of 1.6 standard litres per minute. Tetracycline hydrochloride (Sigma-Aldrich, Munich, Germany) was used as a standard.

**Statistics and reproducibility**. The investigators were not blinded. In the RNAseq experiment, four biological replicates were used because we were interested in the gene clusters that were most strongly induced. In the food choice experiments, the number of animals per arena was limited to 8–20 (depending on the species) due to the requirement that the animals not consume all of the food before the end of the experiment. Statistical tests were deemed unnecessary in certain food choice experiments (Fig. 3, Supplementary Fig. 5) and in the stimulation of aurofusarin synthesis by grazing (Fig. 2d, e, Fig. 6c) owing to the high magnitude of the effects. In time series shown in Fig. 3, Fig. 4b, Fig. 5b, Fig. 8b–e, and Supplementary Fig. 5, the same samples were measured repeatedly. In all other experiments each measurement was taken from a distinct sample.

Bar graphs show means ± SEM, and line graphs show means ± 95% CI. Box plots show medians and lower and quartiles (Q1 and Q3), with whiskers showing the largest (smallest) observation or 1.5-fold of the interquartile range, whichever is smaller (larger) (Q1−1.5 × (Q3−Q1), Q3 + 1.5 × (Q3−Q1). The significance of the differences between the means for the weight gain of mealworms and for the consumption of wheat flour with/without aurofusarin was tested via unpaired *t* tests. The significance of differences in the transcription of the aurofusarin gene cluster was determined as follows: Ct values for target genes and two internal reference genes (glyceraldehyde-3-phosphate dehydrogenase and elongation factor 1a) were adjusted for amplification efficiencies, which were obtained from dilution series[62]. ΔCt was calculated for predated cultures and controls, and the two groups were compared by unpaired two-tailed *t* tests. The induction of gene expression was regarded as significant when relative mRNA levels of at least three genes increased at least threefold and the increase was statistically significant with both reference genes (*0.05 < P < 0.01, **0.01 < P < 0.0001, ***P < 0.0001). Sample size and p values are shown in Supplementary Data File 3.

**Reporting summary**. Further information on research design is available in the Nature Research Reporting Summary linked to this article.

## Data availability

RNAseq data are accessible at ArrayExpress under accession code E-MTAB-6939 [https://www.ebi.ac.uk/arrayexpress/experiments/E-MTAB-6939/] and their analysis is shown in Supplementary Data 1. The authors declare that all other data supporting the findings of this study are available within the article and its Supplementary Information, Supplementary Data files and in the Source Data file.

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

## Acknowledgements

We thank R. Pilot (University of Goettingen) for her excellent technical assistance, F. Gremmes (University of Goettingen) for the construction of a cutting tool, J. Straus (BOKU Vienna, Austria) for advice on experiments with flour, and a referee for pointing out that high ligand concentrations may saturate low-affinity receptors. We are obliged to L. Ruess (Humboldt University Berlin, Germany), A. von Tiedemann (University of Göttingen, Germany), J. Hu (Zhejiang Agriculture and Forestry University, China), T. Miedaner (University of Hohenheim, Germany), H.-W. Dehne (University of Bonn, Germany), R. Geisen (Max-Rubner-Institute, Germany), T. Yli-Mattila (Turku University, Finland), and R. Dastjerdi (Seed and Plant Improvement Institute, Karaj, Iran) for providing us with fungal strains and invertebrate cultures and B.J. van Rensburg (ARC South Africa) for a photograph of moulded maize ear. This work was supported by the German Academic Exchange Service; the German Research Foundation (DFG IRTG 2172); the National Natural Science Foundation of China (21876152); the China Scholarship Concil; and the Ministry for Science and Culture of Lower Saxony, Germany.

## Author contributions

P.K. conceived of and guided the study and wrote the manuscript. Y.X. carried out the interaction experiments as well as the RT qPCR- and HPLC-DAD analysis. M.V. carried out RNAseq analysis, and A.A. purified the aurofusarin and carried out the HPLC-MS- and HPLC-ELSD analysis. L.S. carried out the feeding experiments and RT qPCR with nematodes, K.P. developed and guided the HPLC analysis, and M.R. provided conceptual support for experiments with animals. W.S. generated fungal strains with disrupted biosynthetic pathways, and W.C. carried out the cell culture experiments and provided guidance. All authors approved the manuscript.

## Additional information

**Competing interests:** The authors declare no competing interests.

