## [Peer Review File · Nature Communications]

Reviewers' comments:

Reviewer #1 (Remarks to the Author):

The manuscript by Xu et al. sheds light onto the chemical defense of filamentous fungi against predators. The authors refer to the unresolved issue that most attempts to correlate the biosynthesis of toxic secondary metabolites with deterral of fungal predators were not successful. Based on their results, they put forward the idea that in terms of antifeedant activity, the biosynthesis of large amounts of a specific class of comparably 'nontoxic' pigment is much more effective and used by many filamentous ascomycetes.

The overall conclusions of this manuscript are fascinating and the presented data is convincing. I see the only major flaw of the study in the mechanistic aspects of deterrence and pigment induction. The deterrence mechanism is experimentally not addressed other than by the - negative - toxicity assays. In this regard, it would be very nice if the authors could present lack of toxicity of the purified pigment for another organism besides mealworms for which deterrence is demonstrated; if this is not possible, they could extend their test organisms by *Drosophila melanogaster* larvae both for toxicity and deterrence assays since the pigment could be easily mixed in the food similar to mealworms. In my opinion, however, the (very low) toxicity cannot be the cause for the observed universal deterrence. I rather hypothesize that the pigment either binds, as a signal, to some kind of receptor that is conserved in all tested organisms and this signal recognition leads to the observed avoidance behavior (without toxicity). The large amounts of pigment needed for deterrence might be explained by a low affinity of the pigment for the receptor which, on the other hand, allows binding to the slightly different variants of the same receptor in all tested organisms. Identification of the putative receptor is beyond the scope of this study but the authors could at least formulate some hypotheses in this regard. Alternatively, since the deterring compound is a pigment, it is possible that not the compound itself but rather its color, is perceived by the organisms. I am not really a specialist but I recently read that even nematodes have light-sensing neurons. A possible experiment to distinguish between these two possibilities, could be to replace aurofusarin with other pigments with similar absorption spectra in the food choice experiments. In addition to the deterrence mechanism, the authors should also elaborate a bit more on the mechanism of pigment induction. According to the results in Fig. 2b, aurofusarin production is induced upon physical damage of the mycelium e.g. by chewing and puncturing predators. The authors should mention that in the text and eventually confirm this e.g. as an additional panel of Ext. Data Fig. 3 by local damage of the mycelium on plate with a scalpel. This figure also lacks a negative control of punching in a plastic cylinder without adding any predator.

If the authors can at least partially address above mechanistic issues, I am in favour of accepting the manuscript for publication. Besides above mechanistic issues, the authors should also address some minor issues listed below:

1. Correct title to 'Bis-naphthopyrone pigments protect filamentous ascomycetes from a wide range of predators'
2. Lines 48-50: Sentence is not clear and should be rephrased.
3. Line 63: Explain NLP.
4. Ext. Data Fig. 2: Explain why in panel a FPKM levels instead of relative mRNA levels are shown. Correct 'codifying' to 'coding' in line 634.
5. Line 101: Replace 'amounting to' by 'of'.
6. Line 107: Replace 'incurs' by 'infers'.
7. Fig. 3: It is unfortunate that the picture of the petri dish in panel a shows the reflection of the camera and the researcher. This photograph should be replaced by a better one without such reflections (e.g. by taking off the lid of the plate).
8. Lines 167-172: In order to be able to evaluate the results of these experiments, the authors should provide a proof for the purity of the isolated aurofusarin to exclude that it is contaminated with other mycotoxins. Also, the authors should provide the results of the food choice experiment with aurofusarin-containing wheat flour.

Reviewer #2 (Remarks to the Author):

Overall comments: This paper attempts to show that aurofusarin is an important secondary defense compound in fungi to protect against insect predators. This is a nice study that deserves to be published, but why it belongs in Nature Communications – the authors have not convinced me. To me, this paper belongs in a much more specialized journal. The way it is written is not to engage a general audience and one paragraph of introduction was not enough to grab me and convince me of the larger importance of this story and its greater relevance. In general, the way this paper was written was extremely difficult to follow. This paper requires a reader to have familiarity and specific knowledge of 1. Fungal pathogens 2. RNA seq and transcriptomics 4. Many insect genera AND 5. esoteric secondary defense/chemical compounds. Honestly, I don't know anyone who has enough expertise in all of these areas to be able to assess this paper fully. I am comfortable with fungi and RNAseq to a certain extent, but in order to have a reader understand and follow a paper that spans all of these fields MUCH more effort needs to be put in to explain and define terms and help the reader follow the paper. The paper would benefit greatly from a more expanded and broader introduction and a direct reference to the methods section (which I did not find until after I had read the paper). Also, the figures need to be split up and simplified. I suggest that you pick your favorite point to highlight, and highlight it, and don't try to fit so much into a single figure. I also think many of the statements i.e. "aurofusarin content in fungal mycelia is astonishingly high" sounded rather hyperbolic in absence of citations or comparative data so we can know how high they are relative to other defense compounds.

Line comments:

Line 56: You need to make it clear that you are talking about your experiment now. It would be easier to follow if you said "we sequenced the transcriptome in order to study..." In general, One paragraph of introduction and then going straight into your results is very quick. I would like more introduction and background for why you did this experiment, why did you choose *Fusarium graminearum*? What is the significance of that fungus? Is it just something you happened to have in culture? How many replicates did you have? Is this just one culture? I need more details in the methods here to assess your experimental design.

Figure 1: You need to explain your abbreviations. What is PKS? What is NRPS? There must be a shorter way to list your accession numbers. Also there are many ways to analyze transcriptome data so I would like to see more detail in the supplemental methods or post scripts to github and give us the link.

Line 59-60: You need to introduce these metabolites and why you decided to look for them. I've never heard of aurofusarin before – why is it of interest and how did you know to look for it?

Line 63: this is a lot of jargon and complicated defense agent names. This is seeming very specialized, perhaps belongs in a more specialized journal?

Lines 86-106: In general, an extremely dense paragraph with a TON of jargon. Words I've never heard before reading this paper aurofusarin, dimeric naphtho-gamma-pyrone pigment, those nematode genera you use HPLC abbreviation without ever introducing it first. I can't see that you made a huge effort to make this paper interesting to a more general audience such that you might get with nature communications.

Line 99-101: Why is 1-2% of fungal dry weight "astonishingly high"? What levels were you expecting?

Line 100: Which five *Fusarium* species did you use? Why did you choose them? How many replicates?

Lines 102-104: Not sure why you used ELSD here in addition to the previous HPLC

Line 106-108: To me this sounds like a just so story. I don't think you can say the "extraordinarily high amounts of aurofusarin ...explain why aurofusarin synthesis incurs substantial fitness costs". I think it's more cautious to say these high levels may be a fitness cost, as evidenced by increased growth rates of mutants.." Especially if you didn't measure anything else about the fungus so you don't know what other trade-offs there might be.

Figure 2a. Until you showed me the corn ear I had no idea that *F. graminearum* was a corn pathogen or that was how you were growing it. In general, you need more background, more methods, and more explanation of your experimental set up in order for anyone outside your lab group to be able to follow this story and understand what is happening.

In general figure 2 is extremely dense and includes way too much information for a single figure.

Line 130: are these cultures in liquid culture or are they in corn?

Line 132: why are you switching to the woodlouse when before you were using nematodes?

Line 133: you need to explain this more. Does aurofusarin synthesis cause things to turn red? How would I as a reader know that if you don't tell me?

Line 138: Previously in this paragraph *F.* was used to abbreviate *Fusarium*. I'm assuming the collembolan is another genus but which one? You can't abbreviate like that if you haven't introduced it before.

Line 137-138: How many of each species? Also, how many replicates of each experiment do you have? Did you conduct statistics?

Line 145: this sounds very anecdotal

Line 146: when did you disrupt the biosynthetic pathways for deoxynivalenol and zearalenone and how did you do it?

Line 148: are we back to talking about *Fusarium* with the *F.* now?

Lines 150-151: I don't understand how you can say aurofusarin is the only defense metabolite that has deterred springtails – how many metabolites did you test? Without an exhaustive test I don't see how you can make such statements. Seems hyperbolic to me.

Figure 3. The pictures would be more helpful if you had labels next to them i.e "mealworm" next to the picture of the mealworm for example, to help out those of us who have no idea what a mealworm looks like.

Line 169-170: I fail to see how that experiment answered the question. Please explain further. Also you can't just say "the larvae's strong preference for flour without aurofusarin demonstrated.." without showing data so I can assess the magnitude and effect size

Line 180: Ok so if aurofusarin did not affect meal worm growth, it doesn't seem like it's very toxic to them. How does this help your argument?

Figure 4: you need a legend so we know what the colors mean. I'm guessing it's concentration but

you need a legend to show that.

Line 211: It would be nice if somewhere you defined by a bis-naphthopyrone was

Line 217: I'm not really sure how this paragraph tested whether the findings can be extended to viomellein. Did you try a similar expt to see if insects preferentially ate the fungus without the viomellein versus the fungus with it? Did it simply not work? Or did you not try that test? As written it's unclear what exactly you did.

Line 218-219: A sentence like this requires back up – where are your citations? How can I believe you that these numbers are astonishingly high? What are we comparing to?

Line 221: what is a P450 oxidase?

Line 227-229: citations?

Line 234-235: How do you know that no arthropods have adapted to this defense metabolite? You have tried a handful of insect species, it could be that many species are adapted to it and you just didn't happen to try them.

Line 236: If this is for Nature communications, you need to link this sentence to a larger picture. Why should anyone care about fungal bis-naphthopyrones? What is the greater relevance of this research? I'm not saying it's not important, I'm saying that you need to explain to the audience why it's important and what is the bigger picture.

Line 337: I'm only now seeing that there is a methods section. How come it was not referred to in the main document? Why is it after the references and hidden somewhere I would never find it?

Line 537: 4 biological replicates seems pretty low.

Response to the reviewers' comment on the manuscript "Bis-naphthopyrone pigments protect filamentous ascomycetes from a wide range of predators"

We appreciated the comments of the reviewers, which we found very helpful. The manuscript was revised and new results were added. Originally the manuscript was submitted as a Letter but the editor of [redacted] recommended us direct transfer to Nature Communications, which is why the original submission was not formatted for the latter journal. In the following we respond to the comments of the reviewers.

Reviewer #1 (Remarks to the Author):

Comment: The manuscript by Xu et al. sheds light onto the chemical defense of filamentous fungi against predators. The authors refer to the unresolved issue that most attempts to correlate the biosynthesis of toxic secondary metabolites with deterral of fungal predators were not successful. Based on their results, they put forward the idea that in terms of antifeedant activity, the biosynthesis of large amounts of a specific class of comparably 'nontoxic' pigment is much more effective and used by many filamentous ascomycetes.

The overall conclusions of this manuscript are fascinating and the presented data is convincing.

Response

Thank you for this positive assessment.

*Comment:...I see the only major flaw of the study in the mechanistic aspects of deterrence and pigment induction. The deterrence mechanism is experimentally not addressed other than by the - negative - toxicity assays. In this regard, it would be very nice if the authors could present lack of toxicity of the purified pigment for another organism besides mealworms for which deterrence is demonstrated; if this is not possible, they could extend their test organisms by *Drosophila melanogaster* larvae both for toxicity and deterrence assays since the pigment could be easily mixed in the food similar to mealworms. In my opinion, however, the (very low) toxicity cannot be the cause for the observed universal deterrence.*

Response

We carried out three additional toxicity assays. *D. melanogaster* larvae were fed on food with aurofusarin for two days and allowed to accomplish their development on medium without aurofusarin (Fig. 5b). Aurofusarin has not reduced the number of adults emerging from pupae, showing the lack of developmental toxicity. Our attempt to establish a deterrence assay with

Drosophila larvae failed because boring of larvae into food interfered with counting. Au-rofusarin toxicity was also investigated on the springtail *Folsomia candida* and the isopod *Trichorhina tomentosa*, fed on fungal cultures with and without aurofusarin for 5 weeks (Table 1). The lack of mortality and only slight growth suppression observed in these experiments corroborated our previous results with mealworms and insect cells.

Comment: I rather hypothesize that the pigment either binds, as a signal, to some kind of receptor that is conserved in all tested organisms and this signal recognition leads to the observed avoidance behavior (without toxicity). The large amounts of pigment needed for deterrence might be explained by a low affinity of the pigment for the receptor which, on the other hand, allows binding to the slightly different variants of the same receptor in all tested organisms.

Response

Thank you very much for these ideas. We suggested that large amounts of aurofusarin prevented adaptation, writing "high levels of aurofusarin... may saturate degradation or other activities counteracting the antifeedant effect" (orig. man. L222-223), but we missed the idea of saturating low-affinity receptors. In the new discussion we wrote:

Revision L399-402: "We hypothesise that high levels of aurofusarin in fungal mycelia prevented predators from adaptation by saturating molecular targets of aurofusarin with binding affinities reduced by mutations, or by overwhelming enzymatic degradation. Mycotoxins never accumulate in comparably high concentrations..."

The instructions for authors prevent us from including thanks to referees in the Acknowledgment, but we feel that the saturation of low-affinity receptors was an important idea, and we therefore acknowledged the source.

Comment:....Identification of the putative receptor is beyond the scope of this study but the authors could at least formulate some hypotheses in this regard.

Response

We offered such a hypothesis, based on the wide range of predators that are affected:

Revision L411-415: "The intriguing question for future research is how a single metabolite class deters a wide range of predators. The presence of gustatory receptors triggered by bis-naphthopyrones in phylogenetically distant arthropods, including crustaceans, springtails and insects, indicates that natural ligands of these receptors are compounds common in food substrates of all arthropods, such as proteins or polysaccharides."

Comment: ...Alternatively, since the deterring compound is a pigment, it is possible that not the compound itself but rather its color, is perceived by the organisms. I am not really a specialist but I recently read that even nematodes have light-sensing neurons. A possible experiment to distinguish between these two possibilities, could be to replace aurofusarin with other pigments with similar absorption spectra in the food choice experiments.

Response

To exclude the possibility that the animals recognized and avoided the colour of aurofusarin, we carried out a new food choice experiment with aurofusarin-amended wheat flour in complete darkness (Fig. 4b). Because a single mealworm per arena was used, exposure to dim light for a second was sufficient to locate the animal. The preference for aurofusarin-free flour in the dark was similar to the preference for aurofusarin-free fungal mycelia at ambient light (cf. Fig. 3g).

Comment: ... In addition to the deterrence mechanism, the authors should also elaborate a bit more on the mechanism of pigment induction.

According to the results in Fig. 2b, aurofusarin production is induced upon physical damage of the mycelium e.g. by chewing and puncturing predators. The authors should mention that in the text and eventually confirm this e.g. as an additional panel of Ext. Data Fig. 3 by local damage of the mycelium on plate with a scalpel. This figure also lacks a negative control of punching in a plastic cylinder without adding any predator.

Response

We conducted the suggested experiment, damaging fungal mycelium with an array of razor blades. The results showed that mechanical damage was sufficient to induce aurofusarin synthesis, the synthesis was restricted to the damaged area, and the stimulation of aurofusarin synthesis lasted for at least 120 h (new Fig. 6). These results pointed out at interesting differences between the chemical defence against predators in fungi and plants. We elaborate on these differences in the discussion (revised manuscript L380-389); therefore Fig. 6 was placed into the main manuscript rather than becoming an additional panel in previous Ext. Data Fig. 3 (now Supplementary Fig. 4).

Negative controls for the experiments with plastic cylinders are shown for two fungal species in Supplementary Fig. 4 in panels a and b on the right.

Because shaken cultures of *F. venenatum* accumulated less aurofusarin than still cultures (Orig. Fig 4b; Revised Supplementary Fig. 3), apparently contradicting the hypothesis that mechanical damage triggers aurofusarin synthesis, we injured mycelium of *F. venenatum* by cutting (Supplementary Fig. 6) and observed pigmentation of the mycelia. We described these results in the revised manuscript on L272-277 .

Comment: If the authors can at least partially address above mechanistic issues, I am in favour of accepting the manuscript for publication. Besides above mechanistic issues, the authors should also address some minor issues listed below:

1. Correct title to 'Bis-naphthopyrone pigments protect filamentous ascomycetes from a wide range of predators'

Response: Corrected.

Comment: 2. Lines 48-50: Sentence is not clear and should be rephrased.

Response: The text was rephrased as follows:

Original text, L48-50: "Challenging arthropods with fungal strains with impaired¹⁸⁻¹⁹ and constitutively stimulated²⁰ secondary metabolism generated encouraging results, but the pleiotropic character of the mutations prevented assigning effects to causes."

Revised text, L343-348: "Fungal strains defective in secondary metabolism due to dysfunctional global regulator velvet complex^{20,34}, which controls secondary metabolism and development³⁵, and strains with constitutively stimulated secondary metabolism³⁶ were used in food choice experiments. Predators preferred strains impaired in secondary metabolite synthesis and avoided strains with constitutively stimulated secondary metabolite synthesis, but the pleiotropic character of these mutations prevented identification of metabolites responsible for the effects."

Comment: 3. Line 63: Explain NLP.

Response: The revised text avoids the abbreviation:

Revised manuscript L88-91: "Pathways for the mycotoxins deoxynivalenol and zearalenone – which are toxic to insects^{13,14} – and for necrosis and ethylene-inducing peptide-like proteins – which we hypothesised to be defence agents due to their similarity to lectins²⁴ – were not induced by grazing (Supplementary Data File 1)"

Comment: 4. Ext. Data Fig. 2: Explain why in panel a FPKM levels instead of relative mRNA levels are shown. Correct 'codifying' to 'coding' in line 634.

Response: The genes in 2a encode products of the same kind (peptides). FPKM values reveals differences in mRNA levels among genes, in addition to the effect of grazing, which is useful information. Panels 2b and 2c show pathways with genes encoding products of diverse functions (enzymes, transcription factors, transporters). Comparing mRNA levels among these genes would not be meaningful; therefore relative mRNA levels are shown.

Passage "small secreted cysteine-rich proteins-codifying-genes" was replaced by "genes encoding small secreted cysteine-rich proteins" (caption of Supplementary Fig. 2).

Comment: 5. Line 101: Replace 'amounting to' by 'of'

Response: The sentence was re-phrased (revised manuscript L129)

Comment: 6. Line 107: Replace 'incurs' by 'infers'.

Response: The statement was re-written as follows:

Revised manuscript L396-399: "The maintenance of the high production of aurofusarin that likely infers substantial fitness costs – as evidenced by markedly increased growth rates of mutants with disrupted aurofusarin synthesis^{25, 29} – must have been subjected to a strong selection pressure."

Comment: 7. Fig. 3: It is unfortunate that the picture of the petri dish in panel a shows the reflection of the camera and the researcher. This photograph should be replaced by a better one without such reflections (e.g. by taking off the lid of the plate).

Response: The photograph was replaced by one without reflections.

Comment: 8. Lines 167-172: In order to be able to evaluate the results of these experiments, the authors should provide a proof for the purity of the isolated aurofusarin to exclude that it is contaminated with other mycotoxins.

Response: Due to low solubility of aurofusarin in common solvents, purification of aurofusarin included precipitation by ethanol from melted phenol. This step excluded contamination with known mycotoxins of *F. graminearum* because none of them precipitates in ethanol. The purity of crystallised aurofusarin was verified by HPLC-ELSD (revision L488-489).

Comment:...Also, the authors should provide the results of the food choice experiment with aurofusarin-containing wheat flour.

Response: The results are shown in the new Fig. 4b.

Reviewer #2 (Remarks to the Author):

Overall comments: This paper attempts to show that aurofusarin is an important secondary defense compound in fungi to protect against insect predators. This is a nice study that deserves to be published, but why it belongs in Nature Communications – the authors have not convinced me. To me, this paper belongs in a much more specialized journal. The way it is written is not to engage a general audience and one paragraph of introduction was not enough to grab me and convince me of the larger importance of this story and its greater relevance. In general, the way this paper was written was extremely difficult to follow.

Response: We apologize for the condensed style of our original manuscript, which was formatted for a different journal. The editor of [redacted] recommended us a direct transfer to Nature Communications without format adjustment at that time point. The revised manuscript devotes more space to the background and possesses separated sections dedicate to the results and discussion.

We believe that the manuscript merits publication in Nature Communications because of the conceptual advancements that it provides. Chemical defence in fungi has been studied for decades and several lines of evidence corroborated the defence role of secondary metabolites, but attempts to confirm that fungal defence agents are mycotoxins remained inconclusive. Our works show that rather than mycotoxins, the sought-for defence metabolites are bis-naphthopyrones of low toxicity that are ubiquitous in ascomycetes. The striking features of the new defence mechanism are an unprecedented diversity of predators that are deterred by these metabolites and the lack of adaptation in predators, in spite of persistent exposure. The hypothesis explaining these features by high levels of bis-naphthopyrones in fungal hyphae saturating low-affinity receptors and detoxification activities is a new concept in the biology of fungal defence and in chemical ecology in general. In the original submission we reported our findings on aurofusarin. In the revised manuscript we added new data on two additional bis-naphthopyrones that are produced by many fungal species, corroborating the general validity of the concept. In our opinion, the discovery of a fundamental ecological function of a widespread group of fungal metabolites is important for mycologists, soil ecologists as well as biological chemists, and it is well suited for a wide audience of Nature Communications.

Comment:...This paper requires a reader to have familiarity and specific knowledge of 1.Fungal pathogens 2.RNA seq and transcriptomics 4. Many insect genera AND 5. esoteric secondary defense/chemical compounds. Honestly, I don't know anyone who has enough expertise in all of these areas to be able to assess this paper fully. I am comfortable with fungi and RNAseq to a certain extent, but in order to have a reader understand and follow a paper that spans all of these fields MUCH more effort needs to be put in to explain and define terms and help the reader follow the paper.

Response: We are aware of the challenge posed by a multidisciplinary treatise written by a group of authors with expertise in different fields. If the manuscript is published, the readers will face the same challenge, we therefore appreciate every advice how to present our results intelligibly. In the revised manuscript, we defined all technical terms except those that are generally known. We also provided additional explanations to improve the readability, for instance regarding the use of ELSD.

Comment:...The paper would benefit greatly from a more expanded and broader introduction and a direct reference to the methods section (which I did not find until after I had read the paper). Also, the figures need to be split up and simplified. I suggest that you pick your favorite point to highlight, and highlight it, and don't try to fit so much into a single figure. I also think many of the statements i.e. "aurofusarin content in fungal mycelia is astonishingly high" sounded rather hyperbolic in absence of citations or comparative data so we can know how high they are relative to other defense compounds.

Response: The introduction was expanded and broadened.

Articles in Nature Communications do not provide direct references the methods section, probably to prevent interruptions of the narrative flow. We used the same style but we referred to the methods section explicitly when it was necessary for understanding why a particular method was used (L131).

Fig. 4 was split but we would like to keep Fig. 2 because it allows for direct comparison of values that are related to each other. For instance, the effect of grazing on aurofusarin synthesis is shown at the transcription level on panels 2g,h and at the metabolic level on panels 2d,e. Panels 2c and 2g-j shows mRNA levels for the same genes. Panels 2a,b,f do not show quantitative data, therefore they do not impede comprehension. See also our response to the comment about Fig. 2 on p. 12. The second most complex composite figure is Fig. 3. All panels of Fig. 3 address the food choice of predators; showing these panels next to each other highlights the diversity of predators that are repelled by aurofusarin.

We agree that the statements about the level of aurofusarin in mycelia sounded hyperbolic and would benefit from citations or comparative data. In the revised manuscript, we used more moderate language. We reviewed the literature but have not found any publication reporting comparably high levels of secondary metabolites in fungal mycelia. The manuscript text was revised as follows:

Original text, Abstract: "...causing an accumulation of aurofusarin in fungal mycelia at extraordinary high levels..."

Revised manuscript: the passage was removed

Original text, L218-219: "Aurofusarin content in fungal mycelia is astonishingly high (Figs. 2d-e, 4d), unparalleled by any non-polymeric fungal polyketide...

Revised text, L124-127: "...aurofusarin in grazed mycelia amounted to up to 2.5% of the dry weight (Fig. 2d, e). We were not aware of any non-polymeric secondary metabolite that accumulates in fungal mycelia at such a level..."

Original text, L231-232: "...antifeedants accumulating at unprecedentedly high levels appear to protect fungal mycelia from soil-dwelling predators."

Revised text, L393-394: "Aurofusarin content in *Fusarium graminearum* that has been exposed to predation is very high (Figs. 2d-e)."

Line comments:

Comment: Line 56: You need to make it clear that you are talking about your experiment now. It would be easier to follow if you said "we sequenced the transcriptome in order to study..." In general, One paragraph of introduction and then going straight into your results is very quick.

Response: In the revised manuscript, the passage appears under the heading Results, which shows that we are reporting about our experiment. Thank you for the suggestion of improved wording for the description of RNAseq, we revised the text as follows:

Revised manuscript L80-81: "...we sequenced the transcriptome of the fungus *Fusarium graminearum* that had been exposed to the springtail *Folsomia candida* to reveal which biosynthetic pathways were induced by grazing."

Only a single short paragraph of introduction was allowed in the format of the original manuscript. In the revised manuscript we extended the introduction substantially; see also our response to the first comment on p. 6.

*Comment:...I would like more introduction and background for why you did this experiment, why did you choose *Fusarium graminearum*? What is the significance of that fungus? Is it just something you happened to have in culture? How many replicates did you have? Is this just one culture? I need more details in the methods here to assess your experimental design.*

Response: We have not explained why we have chosen *Fusarium graminearum* because we believe that many filamentous ascomycetes would be suitable. We worked with *F. graminearum* because we had it in the lab, the genome is annotated, and many secondary metabolites have been characterized and assigned to gene clusters, but many fungal species fulfil these conditions. Therefore we have not elaborated on the choice of the species.

The number of replicates in the RNAseq experiment was 4, as shown in captions of Figs. 1 and 2. We will come back to this issue in our response to the last comment below. We used different strains and cultures on solid media and in liquid media, as specified in figure captions and in the description of methods.

Figure 1: You need to explain your abbreviations. What is PKS? What is NRPS? There must be a shorter way to list your accession numbers. Also there are many ways to analyze transcriptome data so I would like to see more detail in the supplemental methods or post scripts to github and give us the link.

Response: We avoided the abbreviations PKS and NRPS in the main text of the revised manuscript and provided explanations when they were needed in the methods.

We shortened the lists of accession numbers by removing prefixes from all except the first accession number for each cluster.

The analysis of the transcriptome is described in the subsection "Transcriptome analysis by RNAseq" of the methods section in the revised manuscript. All scripts that we used are listed here, including version No. and references to original publications, which also provide URLs for the source code. Remote URLs can be found by searching GitHub for the names of the scripts. We have not used custom scripts.

Comment: Line 59-60: You need to introduce these metabolites and why you decided to look for them. I've never heard of aurofusarin before – why is it of interest and how did you know to look for it?

Response: We have not decided to look for these metabolites. On L59-61 of the original manuscript (L84-85 of the revision) we listed the secondary metabolites that were induced by grazing, as revealed by the RNAseq experiment. We could not introduce these metabolites before because we did not know which metabolites would be induced.

Aurofusarin was one of the metabolites induced by grazing. In the original manuscript, the reason why aurofusarin was selected for further work was explained in the subsequent section. In the revised manuscript we provided this explanation immediately. We also removed the names of the other induced metabolites from this passage:

Revised manuscript L63-65: "The biosynthesis pathways for several secondary metabolites were induced via grazing. Among them, bis-naphthopyrone aurofusarin was selected for further investigation as similar metabolites are produced by many fungal species."

Comment: Line 63: this is a lot of jargon and complicated defense agent names. This is seeming very specialized, perhaps belongs in a more specialized journal?

Response: The condensed form of the original manuscript required terse wording. In the revised manuscript, we specify that these products are mycotoxins and explain why they could conceivably act as defence agents. We don't think that reasoning about the role of mycotoxins in fungal defence requires a specialized journal:

Revised manuscript L88-91: "Pathways for the mycotoxins deoxynivalenol and zearalenone – which are toxic to insects^{13, 14} – and for necrosis and ethylene-inducing peptide-like proteins – which we hypothesised to be defence agents due to their similarity to lectins²⁴ – were not induced by grazing (Supplementary Data File 1)."

Comment: Lines 86-106: In general, an extremely dense paragraph with a TON of jargon. Words I've never heard before reading this paper aurofusarin, dimeric naphtho-gamma-pyrone pigment, those nematode genera you use HPLC abbreviation without ever introducing it first. I can't see that you made a huge effort to make this paper interesting to a more general audience such that you might get with nature communications.

Response: We made all efforts to make our work accessible to a general audience. An explanation of the abbreviation HPLC was omitted from the original manuscript because we regarded it as a generally known acronym, and because many recent Nature Communications papers used the abbreviation without explanation. We added the explanation to the revised version of the manuscript (L123).

We fully agree that thus far, aurofusarin was an obscure metabolite known only to specialists. This will certainly change after our work reveals that aurofusarin plays a key ecological role as a defence agent of many fungi against a wide range of predators.

We acknowledge that chemical names may intimidate readers without a background in chemistry. These names, however, provide important information to other readers, and they help understanding structural similarities among the metabolites (Fig. 7).

We agree that the names of the nematodes and other predators will sound obscure to most readers. We have to use accurate species names in the description of our experiments, but no background knowledge on these animals is required. The only information the reader needs is that they can feed on fungi. Our narrative makes clear that the choice of species of nematodes, crustaceans, springtails and insects in our work was not important. The key message is that that aurofusarin and other bis-naphthopyrones (new Fig. 8) repel animal predators belonging to various taxonomic groups.

Comment: Line 99-101: Why is 1-2% of fungal dry weight “astonishingly high”? What levels were you expecting?

Response: We found these levels astonishingly high because secondary metabolites that we knew never reached these levels in mycelia. Most mycotoxins, for instance, accumulate at levels that are two or three orders of magnitude lower. We agree that that the word "astonishingly" is emotional. The statement was revised as follows:

Original manuscript L99-100: "Because these levels were astonishingly high, we determined aurofusarin content in five *Fusarium* species by HPLC..."

Revised manuscript L125-127: "We were not aware of any secondary metabolite that accumulates in fungal mycelia at such a level, and we therefore determined the aurofusarin content in the mycelia..."

See also our response to a similar comment on p. 7-8.

*Comment: Line 100: Which five *Fusarium* species did you use? Why did you choose them? How many replicates?*

Response: The species names are shown in the figure to which the statement refers. We have not provided reasons why these particular species were selected because any *Fusarium* species known to produce aurofusarin could be used. We used species available in the laboratory. The number of replicates is apparent from the figure. The text was revised as follows:

Original manuscript L100-101: "...we determined aurofusarin content in five *Fusarium* species by HPLC with mass spectrometric detection (Fig. 4b)"

Revised manuscript L127-129: "...we therefore determined the aurofusarin content in the mycelia of five *Fusarium* species grown in liquid cultures by HPLC with mass spectrometric detection (HPLC-MS/MS) (Supplementary Fig. 3)."

Comment: Lines 102-104: Not sure why you used ELSD here in addition to the previous HPLC

Response: In the revised manuscript, we explained that we used ELSD because the method did not require aurofusarin standards:

Revised manuscript L129-132: "Because both HPLC-DAD and HPLC-MS rely on aurofusarin standards, which are notoriously unstable (see Methods), extracts of six *F. venenatum* cultures were re-analysed via HPLC with evaporative light scattering

detection (ELSD) for additional verification. ELSD is less accurate than DAD or MS yet does not require aurofusarin standards."

Comment: Line 106-108: To me this sounds like a just so story. I don't think you can say the "extraordinarily high amounts of aurofusarin ... explain why aurofusarin synthesis incurs substantial fitness costs". I think it's more cautious to say these high levels may be a fitness cost, as evidenced by increased growth rates of mutants..” Especially if you didn't measure anything else about the fungus so you don't know what other trade-offs there might be.

Response: Thank you. We revised the text:

Revised manuscript L396-399: "The maintenance of the high production of aurofusarin that may infer substantial fitness costs – as evidenced by markedly increased growth rates of mutants with disrupted aurofusarin synthesis^{25, 29} – must have been subjected to a strong selection pressure."

*Comment: Figure 2a. Until you showed me the corn ear I had no idea that *F. graminearum* was a corn pathogen or that was how you were growing it. In general, you need more background, more methods, and more explanation of your experimental set up in order for anyone outside your lab group to be able to follow this story and understand what is happening.*

Response: We used a picture of an infected corn ear merely to show that red pigmentation caused by aurofusarin can be seen by naked eyes in heavily infected plant material. That *F. graminearum* is a plant pathogen is not relevant for this study. The phenomenon that we studied is unrelated to plant infection and *Aspergillus ochraceus*, which we used as a producer of another defence metabolite similar to aurofusarin (Fig. 8), is not a pathogen and cannot infect living plants. The growth conditions are described briefly in figure captions and in detail in the Methods section.

Comment: In general figure 2 is extremely dense and includes way too much information for a single figure.

Response: We agree that Fig. 2 is dense, but as we explained in our response to a related comment on p. 7, we would like to keep Fig. 2 in order to facilitate comparison of data related to the production of aurofusarin. The upper part does not show quantitative data and the remaining panels consist of merely 7 graphs. Composite figures in Nat. Communication often contain 16 and more graphs, and several such figures often appear in a single article. We did split our original Fig. 4, and the remaining graphs are less complex.

Comment: Line 130: are these cultures in liquid culture or are they in corn?

Response: In the revised manuscript we wrote that the cultures were grown on solid media:

Original manuscript L129-132: "To find out whether aurofusarin synthesis was stimulated by predation in other *Fusarium* species, cultures of *F. poae*, *F. venenatum*, and *F. avenaceum* were subjected to grazing..."

Revised manuscript L135-138: "To determine whether aurofusarin synthesis is stimulated by predation in other *Fusarium* species, cultures of *F. poae*, *F. venenatum*, and *F. avenaceum* on solid media were subjected to grazing..."

Comment: Line 132: why are you switching to the woodlouse when before you were using nematodes?

Response: Both springtails and woodlice were used in this experiment (Supplementary Fig. 4 of the new manuscript). We regarded as important to show that aurofusarin synthesis was stimulated by different predators, which corroborated our hypothesis that aurofusarin and other bis-naphthopyrones protects fungi from a wide range of predators.

Comment: Line 133: you need to explain this more. Does aurofusarin synthesis cause things to turn red? How would I as a reader know that if you don't tell me?

Response: Thank you for pointing this out. We revised the manuscript as follows:

Original manuscript L88-89: "Aurofusarin is a dimeric naphtho- γ -pyrone pigment (Fig. 2f) with a well-characterized biosynthesis²⁵ that colorizes maize cobs infected with *F. graminearum* (Fig. 2a)."

Revised manuscript L113-115: "Aurofusarin is a red pigment known from maize ears infected with *F. graminearum* (Fig. 2a) and pure cultures of the fungus (Fig. 2b). It belongs to dimeric naphtho- γ -pyrones (Fig. 2f)."

Original manuscript L132-133: "Mycelia that had been damaged by predation turned red, indicating that predation stimulates aurofusarin synthesis (Ext. Data Fig. 3)."

Revised manuscript L139-140: "Mycelia of *F. venenatum*, *F. sporotrichioides*, and *F. avenaceum* turned red in areas exposed to predation, indicating that the predation had stimulated aurofusarin synthesis."

Comment: Line 138: Previously in this paragraph *F.* was used to abbreviate *Fusarium*. I'm assuming the collembolan is another genus but which one? You can't abbreviate like that if you haven't introduced it before.

Response: "*F. candida*" was replaced by "*Folsomia candida*" throughout the manuscript. Similarly, "*T. tomentosa*" was replaced by "*Trichorhina tomentosa*".

Comment: Line 137-138: How many of each species? Also, how many replicates of each experiment do you have? Did you conduct statistics?

Response: The number of animals and replicates are specified in the caption of Fig. 3 as well as and in Supplementary File 3. Regarding statistics, Fig. 3 shows 95% confidence intervals for the means but statistical tests of significance of the differences between means were not deemed necessary for these data, as specified in Supplementary File 3.

Comment: Line 145: this sounds very anecdotal

Response: The statement was revised to refer to Fig. 3, which shows the underlying data:

Original manuscript L145: "...after 8 h, most animals were feeding on cultures without aurofusarin."

Revised manuscript L152-153: "... as shown in Fig. 3, after 8 h most animals were feeding on cultures without aurofusarin."

Comment: Line 146: when did you disrupt the biosynthetic pathways for deoxynivalenol and zearalenone and how did you do it?

Response: Both gene disruption mutants are described in the subsection "Fungal strains" of the Methods section:

Revised manuscript L461-464: "The zearalenone-deficient mutant Δ ZEN was generated from the same parent by disrupting the polyketide synthase gene PKS4 involved in zearalenone synthesis. The deoxynivalenol-deficient mutant Δ DON was generated from the strain *F. graminearum* 3211 by disrupting Tri5 gene⁴⁹."

Comment: Line 148: are we back to talking about *Fusarium* with the *F.* now?

Response: In the original manuscript, the abbreviation "*F.*" for *Fusarium* in binomial species names was introduced (L65) shortly after the use of the first binomial (L57). In the revised manuscript, the abbreviation is defined explicitly in the Introduction and used throughout the

manuscript except for subtitles, the first occurrence of genus name *Fusarium* in figures and tables, and the phrase "*Fusarium* species".

Rev. manuscript L62: "... ascomycete *Fusarium graminearum* (*F. graminearum*)."

Comment: Lines 150-151: I don't understand how you can say aurofusarin is the only defence metabolite that has deterred springtails – how many metabolites did you test? Without an exhaustive test I don't see how you can make such statements. Seems hyperbolic to me.

Response: We do not say that aurofusarin is the only defence metabolite that has deterred springtail. We say that reversal of food preference by the disruption of aurofusarin synthesis indicates that aurofusarin was the major or only defence metabolite of *F. graminearum*. The inference is based on the loss of preference for *F. verticillioides* over *F. graminearum* in experiments with *F. graminearum* that was unable to produce aurofusarin. It is conceivable that other metabolites of *F. graminearum* would deter springtails under different conditions, we therefore added "in this experiment":

Original manuscript, L148-151: "The reversal of springtails' food preference for *F. verticillioides* over *F. graminearum* via the disruption of aurofusarin synthesis in *F. graminearum* (cf. Ext. Data Fig. 6a,c with 6b) indicates that aurofusarin had been the major – or only – defence metabolite of *F. graminearum* that had deterred springtails."

Revised manuscript L156-159: "The reversal of the springtails' food preference for *F. verticillioides* over *F. graminearum* via the disruption of aurofusarin synthesis in *F. graminearum* (Supplementary Fig. 5) indicates that aurofusarin had served as the major – or only – defence metabolite of *F. graminearum* deterring the springtails in this experiment."

Comment: Figure 3. The pictures would be more helpful if you had labels next to them i.e "mealworm" next to the picture of the mealworm for example, to help out those of us who have no idea what a mealworm looks like.

Response: The common as well as scientific names of the animals are specified in the caption. The pictures were added to facilitate fast comprehension. We understand that this will be of no use to readers who do not recognize the animals, but adding names next to the pictures would clutter the figure with excessive redundancy. Readers who do not recognize the animals in the pictures will check the caption.

Comment: Line 169-170: I fail to see how that experiment answered the question. Please ex-

plain further. Also you can't just say "the larvae's strong preference for flour without aurofusarin demonstrated.." without showing data so I can assess the magnitude and effect size

Response: We apologize for omitting a reference to Fig. 4c in the original manuscript. In the revised manuscript, Fig. 4c became Fig. 4a and the results of a new experiment with wheat flour are shown in new Fig. 4b. The text was revised accordingly:

Original manuscript L169-172: "To answer this question, mealworms were offered wheat flour amended with aurofusarin and unamended flour in a food choice experiment. The larvae's strong preference for flour without aurofusarin demonstrated that aurofusarin functions as an antifeedant, and that it efficiently deters predators at a concentration similar to its concentration in fungal mycelia."

Revised manuscript L191-197: "To clarify whether indirect effects of the disruption of the aurofusarin pathway may account for the arthropods' preference for fungi in which aurofusarin does not accumulate, mealworms were offered wheat flour amended with purified aurofusarin and unamended flour (Fig. 4). The larvae's strong preference for flour without aurofusarin revealed that aurofusarin possesses antifeedant activity and efficiently deters predators at a concentration similar to its concentration in fungal mycelia upon grazing (Fig. 2d, e)."

Comment: *Line 180: Ok so if aurofusarin did not affect meal worm growth, it doesn't seem like it's very toxic to them. How does this help your argument?*

Response: In addition to this experiment, new experiments confirming low toxicity of aurofusarin in arthropods are shown in the revised manuscript. We regard the low toxicity of defence metabolite aurofusarin as a key result of the study for two reasons. Firstly, it explains why previous efforts to identify defence metabolites among mycotoxins failed. Secondly, we hypothesize that the low toxicity of aurofusarin allows accumulation in mycelia at high levels without self-poisoning, which prevented predators' adaptation:

Revised manuscript, Abstract (L30-31): "Toxicity exerted by aurofusarin in mealworms, springtails, isopods, Drosophila, and insect cells was low, contradicting the previous view that fungal defence metabolites are toxic."

Revised manuscript, L245-246: "The low toxicity of aurofusarin contradicts the hypothesis that fungal defence metabolites are toxic to predators^{12-15,19-20}."

Revised manuscript L399-406: "We hypothesize that high levels of aurofusarin in fungal mycelia prevented predators from adaptation by saturating molecular targets of aurofusarin with binding affinities reduced by mutations, or by overwhelming enzymatic degradation. Mycotoxins never accumulate in comparably high concentrations, which

would probably cause self-poisoning that even protection mechanisms of mycotoxin producers⁴⁵ could not prevent. The synthesis of an antifeedant of low toxicity in large amounts exemplifies a new concept in fungal chemical defence, with aurofusarin, viomellein, and xanthomegnin serving as the first examples."

Revised manuscript L409-411: "Rather than structurally diverse toxins produced in low amounts, structurally similar antifeedants accumulating at high amounts protect fungi from soil-dwelling predators."

Comment: Figure 4: you need a legend so we know what the colors mean. I'm guessing it's concentration but you need a legend to show that.

Response: Panels from Fig. 4 of the original manuscript were transferred to Fig. 5 and Supplementary Fig.3 of the revised manuscript. Explanations of the coloration were added to the captions of both figures:

Revised manuscript, caption of Fig. 5: "Coloration of box plots and bar graphs indicates aurofusarin concentration in flour."

Revised manuscript, caption of Supplementary Fig.3: "The coloration of symbols indicates aurofusarin concentration in mycelia."

Comment: Line 211: It would be nice if somewhere you defined by a bis-naphthopyrone was

Response: We show the structures of bis-naphthopyrone aurofusarin in Fig. 2f and core structures of other bis-naphthopyrones in Fig. 7. Readers with little chemical background can comprehend the formulae. They will see that the molecules consist of two similar parts, and conclude that these parts are naphthopyrones. Explaining the structure in words to readers without a chemical background would be difficult, and it would require a very long text.

Comment: Line 217: I'm not really sure how this paragraph tested whether the findings can be extended to viomellein. Did you try a similar expt to see if insects preferentially ate the fungus without the viomellein versus the fungus with it? Did it simply not work? Or did you not try that test? As written it's unclear what exactly you did.

Response: These results described on L217 of the original manuscript extended our finding to viomellein merely regarding the induction of the synthesis of the metabolite by grazing. We stated explicitly that food choice has not been investigated. In the revised manuscript, we added new experiments on food choice, which confirmed that both viomellein and another bis-naphthopyrone xanthomegnin acted as antifeedants, too:

Original L217: "Whether viomellein also deters predators remains to be determined."

Revised manuscript, L300-307: "The deterrent effect of xanthomegnin and viomellein on the springtail *Folsomia candida* was tested... ..The results showed that the fungal bis-naphthopyrones viomellein and xanthomegnin are antifeedants that exert effects similar to aurofusarin on the springtail *Folsomia candida*."

Comment: L218-219: A sentence like this requires back up – where are your citations? How can I believe you that these numbers are astonishingly high? What are we comparing to?

Response: We addressed this issue in our response to a similar comment above (page 7). In short, we agree that our statements sounded hyperbolic, and we revised the manuscript using more moderate language. We are not aware of any publication reporting comparably high levels of secondary metabolite in fungal mycelia, which we conveyed in the revised manuscript (further revised passages related to his issue are shown on pages 7 and 8 above):

Revised manuscript L124-129: "...aurofusarin in grazed mycelia amounted to up to 2.5% of the dry weight (Fig. 2d, e). We were not aware of any non-polymeric secondary metabolite that accumulates in fungal mycelia at such a level..."

Comment: Line 221: what is a P450 oxidase?

Response: The term was removed.

Comment: Line 227-229: citations?

Response: The presence of bis-naphthopyrones in ascomycetes including references was addressed in other parts of the manuscript (see excerpts below). L227-229 of the original manuscript (L406-407 of the revised manuscript) is condensed because it is part of the conclusions.

Revised manuscript L111-113: "Aurofusarin was selected for further work because it is produced by many fungal species^{25, 26} and because metabolites of similar structures are produced by many genera of ascomycetes²⁷."

Revised manuscript L291-292: "Dimeric naphthopyrones similar to aurofusarin are produced by many genera of filamentous ascomycetes. Core structures of over 50 such metabolites are shown in Fig. 7."

Revised manuscript, caption of Fig. 7: "The taxonomic affiliation of selected fungal genera that produce dimeric naphthopyrones²⁷ is shown on the left with schematic structures of bis-naphthopyrones on the right."

Comment: Line 234-235: How do you know that no arthropods have adapted to this defense metabolite? You have tried a handful of insect species, it could be that many species are adapted to it and you just didn't happen to try them.

Response: We do not claim that no arthropods have adapted to this defence metabolite. In the statement on L234-235 of the original manuscript we suggested that future research will clarify whether large amounts of aurofusarin accumulating in mycelia prevented the adaptation of arthropods. Our results with selected members of crustaceans, collembolans and insects (rather than a handful of insect species) show that predators that have not adapted to bis-naphthopyrones exist and that the phenomenon is widespread. The relevance of this finding and the validity of the question why these arthropods have not adapted would not be affected by possible future findings of other arthropods that have adapted to bis-naphthopyrones.

Comment: Line 236: If this is for Nature communications, you need to link this sentence to a larger picture. Why should anyone care about fungal bis-naphthopyrones? What is the greater relevance of this research? I'm not saying it's not important, I'm saying that you need to explain to the audience why it's important and what is the bigger picture.

Response: We apologise for the condensed form of the original manuscript, which was not written for Nature Communications but was transferred from another journal. With twice as much space for the text, integrating numerous improvements suggested by the reviewers and a dedicated Discussion section we hope that the revised manuscript convincingly conveys the relevance of our work to the reader. We believe that the discovery that metabolites produced by a large number of fungi deter arthropods across a wide spectrum of phylogenies is of fundamental importance for fungal biology and soil ecology, and that the audience of Nature Communications will comprehend and acknowledge its impact.

Line 337: I'm only now seeing that there is a methods section. How come it was not referred to in the main document? Why is it after the references and hidden somewhere I would never find it?

Response: The original manuscript was formatted for a different journal and directly transferred to Nature Communications. The methods section is still not referred to in the main document, which we adopted from the style of most Nature Communication articles, but in the revised manuscript the methods section appears before the references.

Comment: Line 537: 4 biological replicates seems pretty low.

Response: Due to the large sampling depth of RNA sequencing by NGS, a relatively low number of biological replicates are analysed, especially in investigations focusing on strong

effects. Four biological replicates are common. Below we provide examples of recent publications with RNAseq data for three biological replicates. Even two replicates can be used when statistical tests relying on variance are deemed unnecessary, as shown in two additional examples from Nature Communications.

RNAseq with 3 biological replicates

Agostini, R. B. *et al.* Long-lasting primed state in maize plants: salicylic acid and steroid signaling pathways as key players in the early activation of immune responses in silks. *Mol. Plant Microbe Interact.* **32**, 95–106 (2019).

Sang, A. *et al.* Innate and adaptive signals enhance differentiation and expansion of dual-antibody autoreactive B cells in lupus. *Nature Communications* **9**, 3973 (2018).

RNAseq with 2 biological replicates

Hernández, B. *et al.* A virus-encoded type I interferon decoy receptor enables evasion of host immunity through cell-surface binding. *Nature Communications* **9**, 5440 (2018).

Janečka, J. E. *et al.* Horse Y chromosome assembly displays unique evolutionary features and putative stallion fertility genes. *Nature Communications* **9**, 2945 (2018).

REVIEWERS' COMMENTS:

Reviewer #1 (Remarks to the Author):

All issues raised in the original review have been satisfyingly addressed. Thus, I recommend acceptance of the revised manuscript for publication in Nat Comm upon addressing one last minor issue:

Line 32: Delete 'in filamentous fungi' since it implies that the pigments are also common in basidiomycetes, which I conclude from the introduction and discussion, is not the case. In case these pigments also occur frequently in basidiomycetes, this should be clarified in the text.

**Response to the reviewers' comment on the manuscript
"Bis-naphthopyrone pigments protect filamentous ascomycetes from a wide
range of predators"**

REVIEWERS' COMMENTS:

Reviewer #1 (Remarks to the Author): All issues raised in the original review have been satisfactorily addressed. Thus, I recommend acceptance of the revised manuscript for publication in Nat Comm upon addressing one last minor issue: Line 32: Delete 'in filamentous fungi' since it implies that the pigments are also common in basidiomycetes, which I conclude from the introduction and discussion, is not the case. In case these pigments also occur frequently in basidiomycetes, this should be clarified in the text.

Response: The phrase "in filamentous fungi" was removed as the passage was restructured.